



# Year-long Buoy-Based Observations of the Air–Sea Transition Zone off the U.S. West Coast

Raghavendra Krishnamurthy[1], Gabriel García Medina[1], Brian Gaudet[1], William I. Gustafson Jr.[1], Evgueni Kassianov[1], Jinliang Liu[1], Rob K. Newsom[1], Lindsay Sheridan[1], Alicia Mahon[1]

[1]Pacific Northwest National Laboratory, Richland, 99352, USA

*Correspondence to*: Raghavendra Krishnamurthy (raghu@pnnl.gov)

**Abstract.**

Two buoys equipped with Doppler lidars owned by the U.S. Department of Energy (DOE) were deployed off the coast of California in fall of 2020 by Pacific Northwest National Laboratory. The buoys collected data for an entire annual cycle at two offshore locations proposed for offshore wind development by the Bureau of Ocean Energy Management. One of the buoys was deployed approximately 50 km off the coast near Morro Bay in central California in 1100 m of water. The second buoy was deployed approximately 40 km off Humboldt County in northern California in 625 m of water. The buoys provided the first-ever continuous measurements of the air–sea transition zone off the coast of California. The atmospheric and oceanographic characteristics of the area and estimates of annual energy production at both the Morro Bay and Humboldt Wind Energy Areas show that both locations have a high wind energy yield and are prime locations for future floating offshore wind turbines. This article provides a description and comprehensive analysis of the data collected by the buoys is conducted and a final post-processed dataset is uploaded to a data archive maintained by DOE. Additional analysis was conducted to show the value of the data collected by the DOE buoys. All post-processed data from this study are currently available on the Wind Data Hub website, https://a2e.energy.gov/data#. Near-surface, wave, current, and cloud datasets for Humboldt and Morro Bay are provided at 10.21947/1783807 and 10.21947/1959715, respectively. Lidar datasets for Humboldt and Morro Bay are provided at 10.21947/1783809 and 10.21947/1959721, respectively.

## 1 Introduction

The Biden Administration has announced a national goal in the United States to deploy 15 GW of floating offshore wind energy by 2035, much of which will be off the coast of California. Approximately two-thirds of our nation's offshore wind potential is located over areas with waters too deep for traditional, fixed-bottom offshore wind foundations, instead requiring floating platforms. However, floating offshore wind technology is still maturing and costs 50% more than fixed-bottom technologies. The U.S. aims to reduce the levelized cost of energy of floating offshore wind by 45% by 2030. Cost reductions



are possible with increased offshore data collection, using oceanographic buoys to better understand meteorological and oceanographic conditions, particularly wind speed and direction at hub-height, where offshore turbine blades will be spinning.

Offshore data are used for wind model validation and forecasting which allows wind developers and consultants the ability to predict and quantify power production and turbine loads and support finance and investment decisions.

The Bureau of Ocean Energy Management delineated two offshore Wind Energy Areas (WEAs) near Humboldt and Morro Bay. These two areas are expected to support most of the offshore wind energy development over the coast of California (Dvorak et al., 2010, Musial et al., 2016). Along the U.S. West Coast, the impact of atmospheric and oceanographic conditions

on floating offshore wind turbines is largely unknown. Due to the sharp gradient in the bathymetry of the seafloor extending from the coastline, future wind farms will primarily be composed of floating offshore wind turbines. Furthermore, the accuracy of existing high-resolution coupled ocean–atmosphere models in estimating the wind resource is questionable because of the complex wind–wave–terrain interactions, extensive cloudiness, and shallow atmospheric boundary layers typically observed in this region. Recent surface buoy climatological analysis using National Data Buoy Center (NDBC) buoys along the

California coast showed seasonal and diurnal variability observed at several sites (Wang et al., 2019). So far, to the best of the authors' knowledge, there have been no wind observations for 1-year within the air-sea transition zone (ASTZ, encompassing the upper oceanic boundary layer and lower marine atmospheric boundary layer) collected over an annual cycle off the coast of California. Observing the ASTZ is aimed to improve our understanding of the ocean-atmosphere coupled processes which influence the atmospheric dynamics and climate change patterns across many regions of the globe. Certain

processes that the ASTZ influences within the California region are atmospheric rivers, shallow boundary layers, droughts, hurricanes, Pacific tropic Coral loss and several sub-seasonal-to-seasonal time scale processes (Armstrong McKay et al., 2022).

Pacific Northwest National Laboratory (PNNL) operates two lidar buoys on behalf of the U.S. Department of Energy (DOE) in areas targeted for offshore wind development. The buoys are collecting first-of-its-kind publicly available, multi-seasonal hub-height observations (Gorton et al., 2020, Krishnamurthy et al., 2021). To estimate the annual wind resource at

the two potential development areas in California, the two DOE buoys, equipped with Doppler lidars and a suite of meteorological and oceanographic instrumentation, were deployed at those locations for a year. One of the buoys (Buoy #130) was deployed approximately 50 km off the coast near Morro Bay in central California in 1100 m of water. The second buoy (Buoy #120) was deployed approximately 40 km off Humboldt County in northern California in 625 m of water. The resulting freely available data provide wind farm developers with critical information on the available wind resource at these locations.

Buoy data can be freely accessed through the DOE-funded Wind Data Hub (formerly the Atmosphere to Electrons Data Archive and Portal; https://a2e.energy.gov/data).

One of the DOE lidar buoys was initially deployed off the coast of Virginia in 2015, and the other buoy was first deployed off the coast of New Jersey in 2016. The data from the buoys at these locations provided the first open-ocean, hub-height wind resource characterizations over a full annual cycle at hub height in the United States. Prior analysis of the buoy data collected

off the U.S. East Coast provided the experience to inform both instrument configurations and performance of various algorithms on current DOE buoy instrumentation (Shaw et al., 2020). The quality control procedures are an important aspect



of the buoy data and are currently also being investigated by International Energy Agency Task 43, expanding a wind resource assessment data model for floating lidars. In this article, substantial analysis of the data collected from the two buoys operated off the coast of California is presented. Section 2 provides details of the buoy instrumentation, lidar validation study, and deployment details. Section 3 provides an assessment of the overall data availability, quality control checks applied to the data, and algorithms used to post-process the buoy data. Post-processing algorithms were applied to data from the Doppler lidar, wave sensor, current profiler, and pyranometer data.  Section 4 provides a climatological analysis of winds near the surface together with thermodynamic variables measured at the surface for the deployment periods at both Morro Bay and Humboldt. Detailed annual analysis of the Doppler lidar winds and turbulence, and of oceanographic observations regarding sea state and cloud distributions at both deployments, are also presented. Beyond the buoy observation analyses, we have also made a preliminary investigation of the wind profiles in the context of classical Monin–Obukhov (MO) similarity theory of the atmospheric surface layer. Section 5 provides details of the code and data availability. Finally, Section 6 provides a summary of all the observations.

## 2     Buoy instrumentation, validations, and deployment

### 2.1     Instrumentation

The DOE buoys used in this study have state-of-the-art instrumentation to measure the offshore wind resource. The buoys were procured as AXYS WindSentinel™ buoys in 2014 but have been significantly altered and upgraded since their initial procurement. The buoy hulls are identical to those of Navy Oceanographic Meteorological Automatic Device (NOMAD) buoys, which are known to be durable and to have good performance characteristics. They are also the principal hulls that were used on NDBC stations deployed off U.S. coasts prior to the shift in use to discus buoys. The DOE buoys have aluminium boat-shaped hulls (see Figure 1 below) that are 6.1 m (20 ft) long and 3.0 m (9.8 ft) wide with a depth of 2.5 m (7.0 ft). A stainless-steel mooring yoke holds the buoy to its mooring. The yoke allows the buoy to rotate about the pitch axis but prevents the buoy from roll rotation (Timpe and Van de Voorde 1995).

The mast on the bow of the buoy supports a satellite antenna, navigation lights, an AIS GPS/VHF antenna, a cup-and-vane anemometer (4.1 m ASL), an ultrasonic anemometer (4.1 m ASL), an air temperature and relative humidity sensor (3.7 m ASL), and a solar radiation sensor (4 m ASL). A radar reflector is placed at the stern of the buoy, and a wind profiler is placed mid-buoy deck that captures winds from 40 m to 250 m ASL. In addition to atmospheric instruments, the buoy supports several oceanographic measurements, including sea-surface temperature measurements, an acoustic Doppler current profiler that provides ocean current speed and direction from the surface to 200 m water depth, a directional wave sensor, sea conductivity, and multiple inertial motion units to register buoy movements necessary for accurate wind calculations. Table 1 lists all the instruments, their make/model, and measurements provided by the DOE buoy during the California deployment (for more details on instruments, see Severy et al., 2020). The instrumentation on both buoys was



identical. Deployment and maintenance of the buoys and instrumentation, as well as some post processing, was performed by AXYS Technologies through a subcontract from PNNL.

95

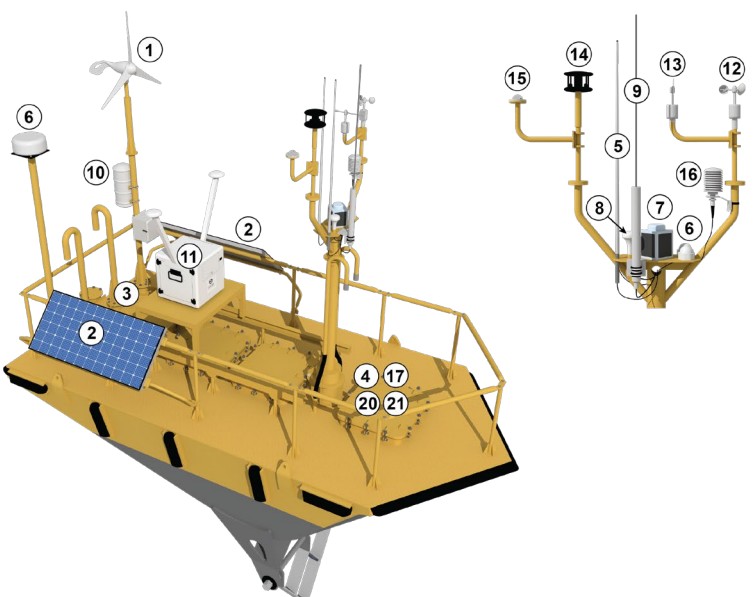

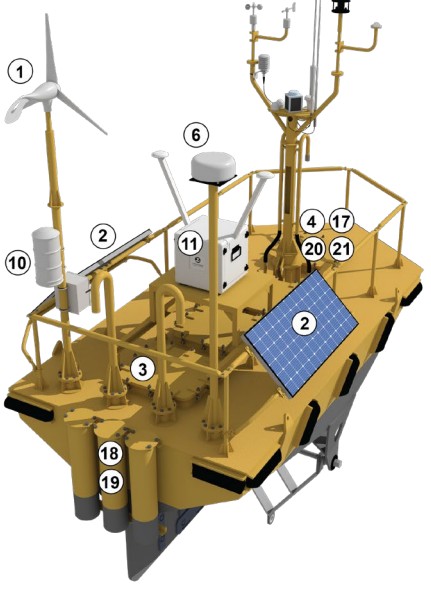

**Power, Data, Communication, & Navigation**

1. Turbine
2. Solar panels
3. Diesel generator (compartment)
4. Data loggers (compartment)
5. Cellular antenna
6. Satellite antenna
7. Navigation light
8. AIS GPS antenna
9. AIS VHF antenna
10. Radar reflector

**Meteorological**

11. Wind profile
12. Wind speed (cup anemometer)
13. Wind direction
14. Wind speed & direction (ultrasonic anemometer)
15. Solar radiation
16. Air temperature & relative humidity
17. Barometric pressure (compartment)

**Oceanographic**

18. Water velocity profile (moonpool)
19. Salinity & water temperature (moonpool)
20. Wave spectrum (compartment)
21. Water temperature (compartment)

**Figure 1: 3-Dimensional schematic of the DOE buoy and sensor placement.**

**Table 1. Description of instrument manufacturer and models.**

| Sensor Type | Make/Model | Measurements |
|---|---|---|
| Wind profiling lidar with built-in inertial motion unit (IMU) | Leosphere/Windcube 866 | Vertical profile of motion-compensated wind speed and direction using the internal IMU, wind dispersion, and spectral width |
| Cup anemometer | Vector Instruments/A100R | Horizontal wind speed, near surface |



| Wind vane | Vector Instruments/WP200 | Horizontal wind direction, near surface |
|---|---|---|
| Ultrasonic anemometer | Gill/WindSonic | 2D wind velocity and direction, near surface |
| Pyranometer | Licor/LI-200 | Global solar radiation |
| Temperature | Rotronic/MP101A | Air temperature |
| Relative humidity | Rotronic/MP101A | Relative humidity |
| Acoustic Doppler current profiler | Nortek/Signature 250 | Ocean current speed and direction from sea surface to 200 m water depth |
| Conductivity temperature depth (CTD) | Seabird/SBE 37SMP-1j-2-3c | Conductivity and sea surface temperature |
| Directional wave sensor | AXYS/TRIAXYS NW II | Directional wave spectra, wave height, and wave period |
| Water temperature | AXYS/YSI | Sea surface temperature |
| IMU for wind vane correction | MicroStrain/3DM GX3 25 | Yaw, pitch, roll, and global position |
| Additional IMU for lidar motion compensation (underneath the lidar) | MicroStrain/3DM GX5 45 | Yaw, pitch, roll, linear velocity, global position, magnetometer, and gyroscope |

## 2.2    Instrument calibration and validation

The three inertial measurement units (IMUs) onboard the buoy were calibrated through a swing test where the buoy was swung multiple times near the shore using a crane.  All the IMUs recorded similar roll, pitch, and yaw measurements at different temporal resolutions.  The GX3-25 measured at 1 Hz, while the GX5-45 and Windcube in-built IMU measured at 10 Hz.  The GX5-45 also measured the linear velocity, and angular acceleration. The GX5-45 IMU data is used for motion-compensating the lidar wind speed, direction, and turbulence measurements. Before the California deployment, independent performance verification of both the floating lidars was conducted at the Martha's Vineyard Coastal Observatory by DNV GL Energy USA Inc. (DNV GL). This verification was performed against a fixed industry-accepted reference lidar. Wind speed and wind direction were compared against corresponding key performance indicators and acceptance criteria using the method provided in the Roadmap toward Commercial Acceptance (CarbonTrust 2018). In summary, both the lidars (Buoy #120 and #130) demonstrated their ability to produce accurate wind speed and direction data. The lidar wind speed uncertainties were calculated to be less than 2%, and correlation coefficients against the reference lidar wind speeds were greater than 99%. A summary of the validation can be found in Gorton et al. (2020), and the validation report is available public on the PNNL buoy webpage.[1]

---

[1] https://www.pnnl.gov/projects/lidar-buoy-program/technical-specifications



## 2.3    Field deployment summary

Figure 2a shows the location of the two buoys deployed within the Morro Bay and Humboldt WEAs. Multiple NDBC buoys are also located several kilometres away from the buoy sites. The NDBC buoy data are a good reference to confirm that the DOE buoy data are consistent in near-surface atmospheric and oceanographic variables. The spatial variability of the atmospheric and oceanographic variables can be assessed by comparing the measurements from these stations. The Morro Bay buoy (Buoy #130) was deployed offshore from September 28, 2020, to October 16, 2021. Before the deployment, all the onboard IMU sensors were calibrated using a swing test, but significant drift was observed in the internal IMU within the lidar. All the IMUs recorded similar trends in pitch, roll, and yaw with no time delay observed. The buoy was towed and moored approximately 50 km off the coast at approximately 35.71074° N and 121.84606° W. The buoy was deployed at 1050 m of water depth. The excursion radius was 1256 m with a mooring length of ~1640 m. Figure 2c shows the final deployment picture of the Morro Bay buoy. At Humboldt, the buoy (Buoy #120) was deployed offshore on October 8, 2020. The buoy was towed and moored approximately 40 km off the coast at approximately 40.9708° N and 124.5901° W. The buoy was deployed at a water depth of 575 m with a mooring length of 1050 m and an excursion radius of ~800 m. Figure 2b shows the final deployment picture of the Morro Bay buoy. The raw and averaged data from the buoy were sent in near real time to the Wind Data Hub each day.

Figure 3 shows the instrument uptime for various sensors onboard both the buoys and Table 2 shows the cumulative campaign data availability for each sensor. Overall, the uptime of the buoy at Morro Bay was ~98% and at Humboldt was 91%. Sensors onboard the Morro Bay buoy performed adequately throughout the deployment and did not need a service visit or intervention. At Humboldt, the buoy unfortunately suffered some data loss due to challenging weather conditions. On December 8, 2020, the buoy encountered a large wave event off Humboldt that resulted in damage to the buoy's power systems. To avoid additional issues, the buoy was remotely shut down until it was recovered for repair. Due to unfavourable weather conditions and other unforeseen delays, the buoy was re-deployed on May 24, 2021, at 20:30 UTC. The buoy continued to experience power system issues which were ultimately resolved during a service visit on April 9, 2022. The Doppler lidar data had spotty availability during this period because the lidar was turned on only during forecasts with high winds, i.e., when it could be powered solely by renewable sources. Finally, the Humboldt buoy was decommissioned on 28 June 2022 at 1330 UTC.

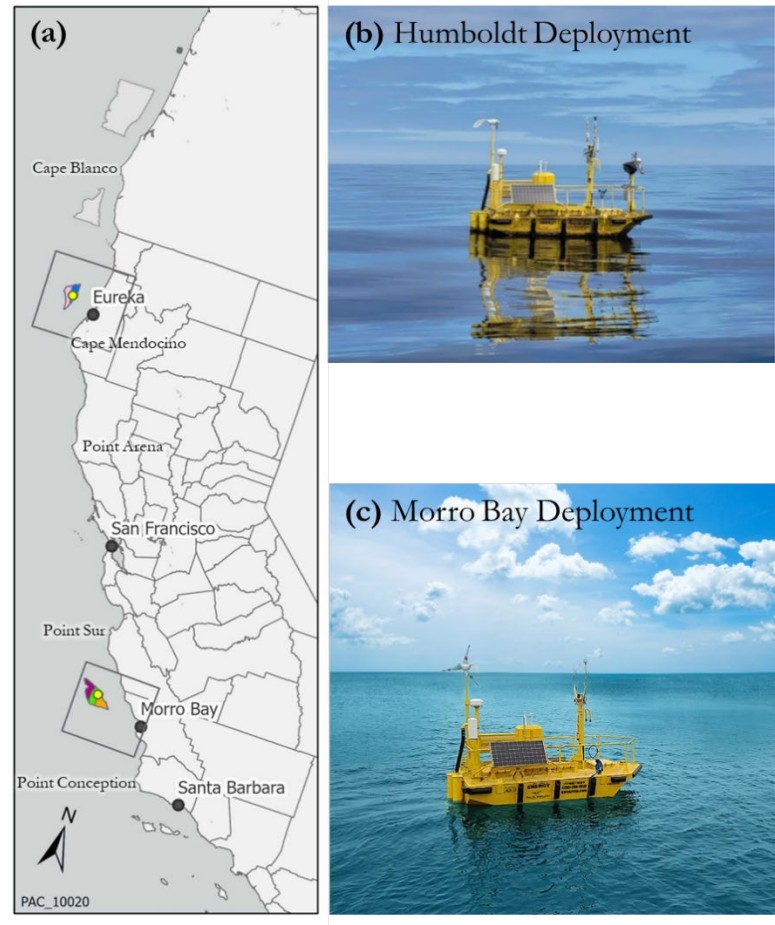

**Figure 2: (a) The yellow circles indicate the location of the two buoys within the California wind energy lease regions (color coded, courtesy of BOEM). (b) A picture of the Humboldt buoy deployment in October 2020. (c) A picture of the Morro Bay buoy deployment in September 2020. Photos courtesy of AXYS Technologies, Inc.**





**Figure 3: (a) Instrument uptime at Morro Bay for various instrument sensors and (b) instrument uptime at Humboldt for various sensors.**

**Table 2. Cumulative campaign data availability after preliminary post-processing of the raw data from each sensor.**

| Sensor | Cumulative campaign system availability for Morro Bay (%) | Cumulative campaign system availability for Humboldt (%) |
|---|---|---|
| Air temperature | 96.64 | 63.92 |
| Relative humidity | 22.83 | 76.96 |
| Barometric pressure | 96.06 | 76.92 |
| Sea-surface temperature | 95.36 | 63.51 |
| Surface winds | 96.65 | 77.14 |
| Wave sensor | 95.98 | 78.41 |
| Ocean currents | 86.08 | 80.61 |





| | | |
|---|---|---|
| Conductivity | 95.16 | 63.51 |
| Pyranometer (solar irradiance) | 95.36 | 63.51 |
| Doppler lidar | 96.55 | 61.22 |

The Doppler lidar was configured to measure at predetermined heights or "range-gates." The height of the lidar window above the mean sea level (MSL) is 2.350 m. Therefore, for the actual measurement height, the range-gate configuration must be added with the height of the lidar window above MSL. The range-gates were configured at 40, 60, 80, 90, 100, 120, 140, 160, 180, 200, 220, and 240 m relative to the lidar window.

## 3    Data analysis

Buoy measurements undergo standard quality checks, such as making sure the sensor is not providing data beyond manufacture limits, detecting abnormal spikes in the data, filtering based on signal-to-noise ratio for the lidars, etc. These automated checks do not necessarily filter all bad data. This section describes the extra data quality checks, which include instrument cross-checking, physics-based analyses, and comparisons with nearby sensors. As a starting point, only data that were collected when the buoy was moored at the target location were considered in this analysis (any measurements collected during towing or services onshore were removed). Although measurements of pressure or temperature are valid when the buoy is moving, we mask them because they do not necessarily represent the conditions at the deployment location. Filtering by watch circle is performed in addition to instrument malfunction, the extent of the watch circle for both deployments is shown in Figure 4. During the Morro Bay deployment, 490 measurements were flagged as bad, no measurement as questionable, and 52,083 as good. During the Humboldt deployment, 4,461 were flagged as bad, 2 as questionable, and 68,312 as good. The two questionable measurements occurred when the buoy drifted a few tens of meters outside the watch circle but reported data within the watch circle during the previous and next measurements. The location data is also available within the surface data. A consolidated list of variables available in the post processed data is provided in Appendix B.

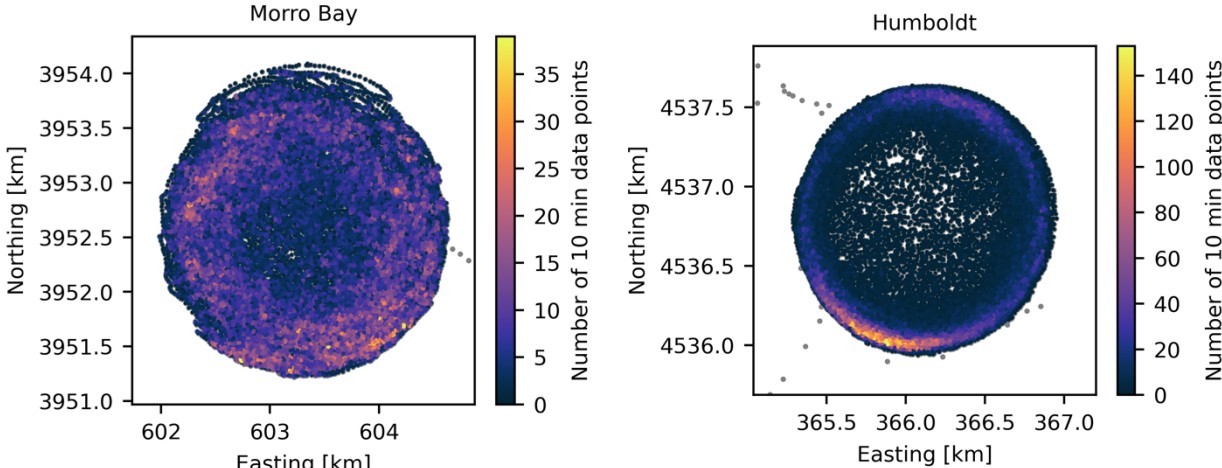

**Figure 4: Location of lidar buoys during the deployments. Gray points have been identified as bad data. The color scale indicates the number of measurements in a 50 square meter area. Data are projected in Universal Transverse Mercator zone 10.**

**3.1    Surface meteorological data processing and filtering**

Each surface measurement (wind speed, wind direction, pressure, air temperature, and relative humidity) in the processed 10-minute surface meteorological dataset was subjected to the following levels of quality analysis and filtration (Krishnamurthy and Sheridan et al., 2023a, 2023c). First, if no instrument aboard the buoy (including the lidar, surface meteorological, and oceanic instruments) was reporting for a given timestamp, the event was considered a power outage, all surface measurements

were assigned a value of NaN, and all surface measurement codes were set to 2. Second, if an individual surface instrument was not reporting for a given timestamp, but other surface instruments were reporting, the event was deemed an instrument failure and the individual surface measurement was assigned a value of NaN and a code of 3. Third, if no surface measurements were reporting, but the lidar or oceanic instruments were reporting, the event was classified as a communications issue, all surface measurements were assigned a value of NaN, and all surface measurement codes were set to 4. Fourth, individual

surface measurements that were considered incorrect (atypical or unphysical) or outside the watch circle were filtered out by being assigned a value of NaN, and the corresponding individual surface measurement code was set to 5. Examples of atypical or nonphysical data include reported wind speeds less than 0 m s$^{-1}$, wind directions less than 0° or greater than 360°, and relative humidity measurements outside the range of 0% to 100%. Any surface measurements that were physically probable but significantly diverged from nearby observations were assigned a code of 1. All remaining surface measurements were

deemed good and assigned a code of 0.



The data recovery and quality of the surface meteorological observations during the Morro Bay deployment was high for all variables except relative humidity, with more than 90% of the data designated as good (Table 3). Due to instrument failure, only 22.7% of the relative humidity observations were deemed usable. For all surface meteorological variables, power outages, communications issues, and watch circle/incorrect data filtration affected the data recovery and quality for 0.8%, 4.2%, and 0.6% of the Morro Bay deployment, respectively. Corrected near-surface wind directions were provided by AXYS and utilized in the Morro Bay near-surface b0 dataset.

For the Humboldt deployment (Table 4), the data recovery and quality of the surface meteorological observations were both lower than at Morro Bay, even outside the extensive power outage periods discussed in Section 2.3. All surface instruments during the Humboldt deployment were subject to power outages for 18.3% and communication issues for 2.5%. Except for air temperature, the data recovery and quality for all surface meteorological variables were both affected by instrument failure for 0.1% or less of the Humboldt deployment and by watch circle/incorrect data filtration for 6.7% of the Humboldt deployment. No wind speed, wind direction, or pressure data was flagged as suspect, leaving 72.5% of good data for these variables. Relative humidity observations during the period of 23 February 2022 to 29 April 2022 (9.6% of the Humboldt deployment) were flagged as suspect due to atypical deviations from the nearest NDBC-buoy-derived relative humidity, leaving 62.9% of good data. Missing air temperature observations due to instrument failure occurred during 1.2% of the Humboldt deployment. In addition to the watch circle filtration, air temperature data at Humboldt was also filtered during four periods when the recorded measurements atypically dropped to around −30 °C: 5–20 September 2021, 15–23 November 2021, 13 February 2022, and 24 February–5 March 2022. The total watch circle/incorrect data filtration for air temperature was 11.5% at Humboldt. Air temperatures during the period of 5 March 2022 to 29 April 2022 (8.1% of the Humboldt deployment) were flagged as suspect due to atypical deviations from the nearest NDBC buoy air temperatures, leaving 58.4% of good data.

**Table 3. Surface meteorological data quality flags for the Morro Bay deployment.**

| Deployment | Morro Bay | | | | | |
| --- | --- | --- | --- | --- | --- | --- |
| | 55,152 Possible 10-Minute Data Points | | | | | |
| Data Quality Flag | 0 Good Data | 1 Suspect Data | 2 Incorrect Data | 2 Incorrect Data | 2 Incorrect Data | 2 Incorrect Data |
| Data Quality Code | 0 Good | 1 Suspect | 2 Power Outage | 3 Instrument Failure | 4 Communication Issue | 5 Watch Circle / Incorrect Data Filter |
| Wind Speed (ultrasonic) | 51,042 92.5% | 0 0.0% | 419 0.8% | 1,041 1.9% | 2,294 4.2% | 356 0.6% |



| | | | | | |
|---|---|---|---|---|---|
| Wind Speed (cup) | 52,082 94.4% | 0 0.0% | 419 0.8% | 1 0.0% | 2,294 4.2% | 356 0.6% |
| Wind Direction (ultrasonic) | 49,760 90.2% | 0 0.0% | 419 0.8% | 2,323 4.2% | 2,294 4.2% | 356 0.6% |
| Wind Direction (vane) | 50,783 92.1% | 0 0.0% | 419 0.8% | 1,300 2.4% | 2,294 4.2% | 356 0.6% |
| Pressure | 51,770 93.9% | 0 0.0% | 419 0.8% | 313 0.6% | 2,294 4.2% | 356 0.6% |
| Air Temperature | 52,083 94.4% | 0 0.0% | 419 0.8% | 0 0.0% | 2,294 4.2% | 356 0.6% |
| Relative Humidity | 12,524 22.7% | 0 0.0% | 419 0.8% | 39,559 71.7% | 2,294 4.2% | 356 0.6% |

**Table 4. Surface meteorological data quality flags for the Humboldt deployment.**

| Deployment | Humboldt 91,859 Possible 10-Minute Data Points | | | | | |
|---|---|---|---|---|---|---|
| **Data Quality Flag** | 0 **Good Data** | 1 **Suspect Data** | 2 **Incorrect Data** | 2 **Incorrect Data** | 2 **Incorrect Data** | 2 **Incorrect Data** |
| **Data Quality Code** | 0 **Good** | 1 **Suspect** | 2 **Power Outage** | 3 **Instrument Failure** | 4 **Communication Issue** | 5 **Watch Circle / Incorrect Data Filter** |
| Wind Speed (ultrasonic) | 66,600 72.5% | 0 0.0% | 16,795 18.3% | 19 0.0% | 2,289 2.5% | 6,156 6.7% |
| Wind Speed (cup) | 66,603 72.5% | 0 0.0% | 16,795 18.3% | 20 0.0% | 2,289 2.5% | 6,152 6.7% |
| Wind Direction (ultrasonic) | 66,604 72.5% | 0 0.0% | 16,795 18.3% | 18 0.0% | 2,289 2.5% | 6,153 6.7% |
| Wind Direction (vane) | 66,604 72.5% | 0 0.0% | 16,795 18.3% | 18 0.0% | 2,289 2.5% | 6,153 6.7% |
| Pressure | 66,559 72.5% | 0 0.0% | 16,795 18.3% | 62 0.1% | 2,289 2.5% | 6,154 6.7% |
| Air Temperature | 53,650 58.4% | 7,453 8.1% | 16,795 18.3% | 1,075 1.2% | 2,289 2.5% | 10,597 11.5% |
| Relative Humidity | 57,796 62.9% | 8.808 9.6% | 16,795 18.3% | 19 0.0% | 2,289 2.5% | 6,152 6.7% |



## 3.2 Impact of motion correction on wind and turbulence estimates

Problems with the lidar internal Windcube IMU were known during the initial validation of the lidar. As a result, two backup IMUs were procured and installed on each lidar before the Morro Bay and Humboldt deployments. The backup IMUs were
3DM-GX5-45 (hereon referred to as the GX5) from Microstrain Sensing. These devices were programmed to output platform attitude data at 10 Hz and position and velocity data at 4 Hz. Figure 5 and Figure 6 show comparisons between roll and pitch measurements from the Windcube IMU and the GX5 at the Humboldt and Morro Bay sites, respectively. There is little agreement between the measurements from these two IMUs. The Windcube's measurements exhibit large fluctuations that are not physically realistic. Additionally, the Windcube's pitch and roll measurements show almost no correlation with the GX5
measurements. During the Humboldt and Morro Bay deployments, the Windcube lidars motion compensated wind estimates were affected by the internal IMU drift, which added significant noise to the 1-second motion-corrected wind profiles stored in real-time data files (*.rtd). Because these data are used to derive the final motion-compensated 10-minute statistically averaged data (i.e., the *.STA files), it is important to understand the impact of the bad Windcube internal IMU data on these results. Our approach involved reprocessing the uncorrected wind profiles using attitude data from the backup GX5-45 IMU
unit. This allowed us to evaluate the impact of motion correction on wind speeds and velocity variances. The non-motion-compensated data are also available, which are referred to as the *.stdrtd (1 Hz) and *.stdsta (10-minute averaged) files.





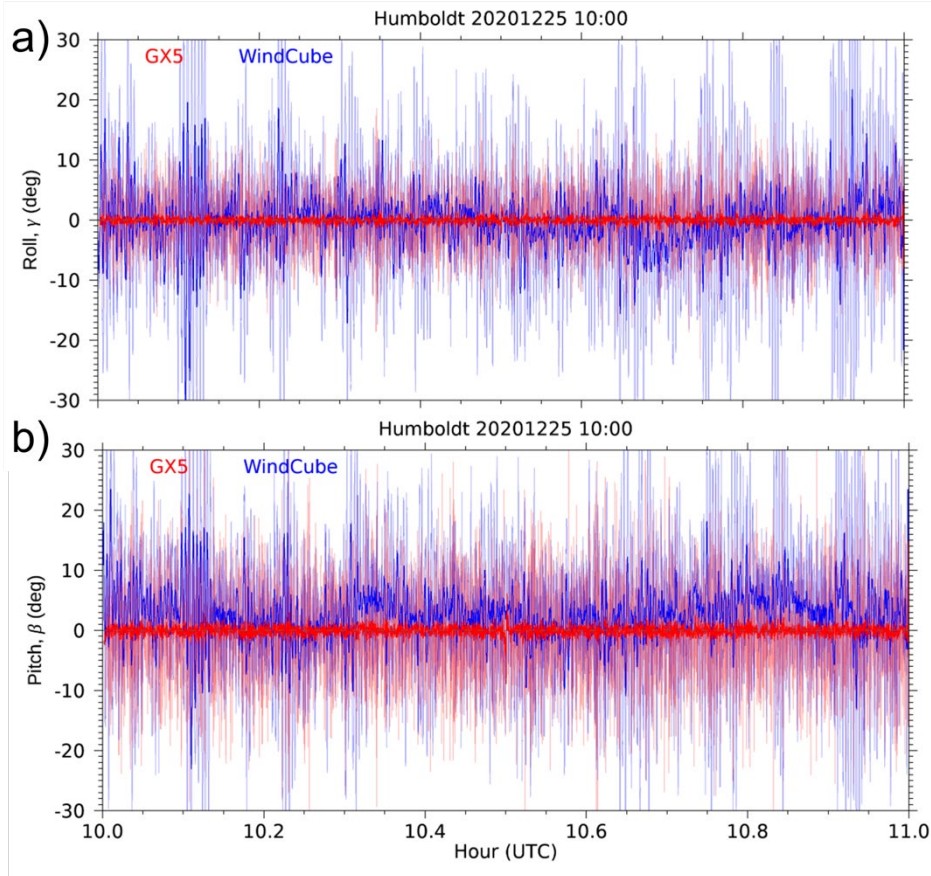

**Figure 5: Comparison between the GX5 (red) and Windcube (blue) IMU measurements of (a) roll and (b) pitch observed on December 25, 2020, at the Humboldt deployment. Lighter shades show 1-second variability, and the darker shades show 10-second box-car averages.**
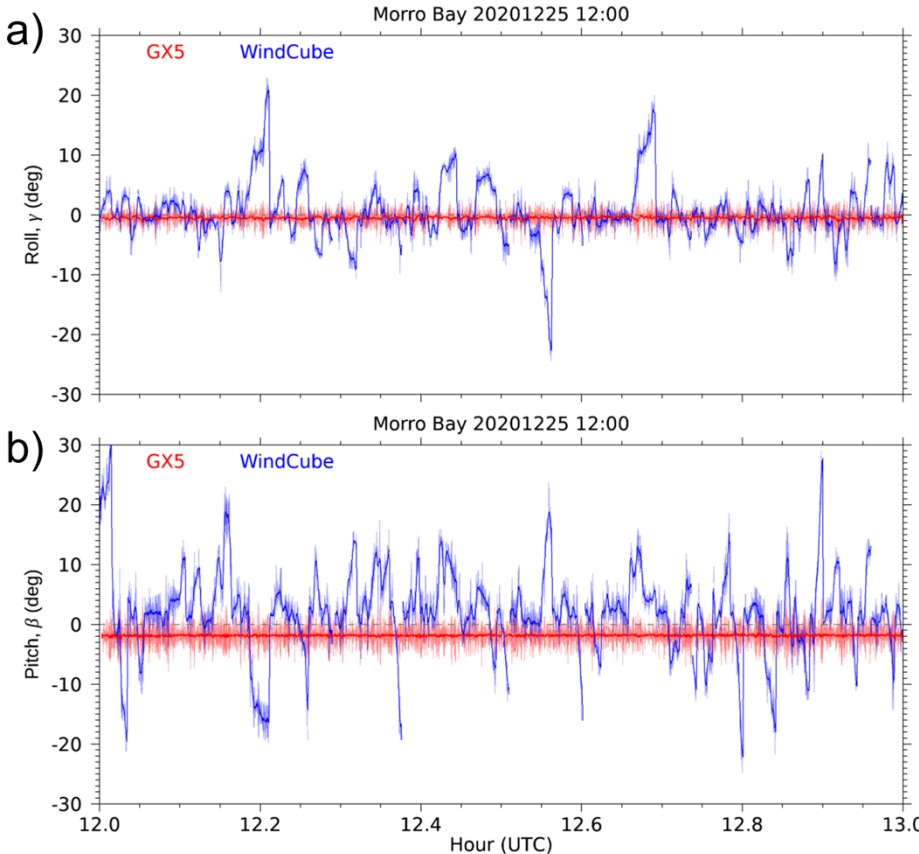

**Figure 6: Comparison between the GX5 (red) and Windcube (blue) IMU measurements of (a) roll and (b) pitch**
**observed on December 25, 2020, at the Morro Bay deployment. Lighter shades show 1-second variability, and the**
**darker shades show 10-second box-car averages.**

The yaw measurements used in the Windcube's motion-correction procedure were derived from a differential GPS (DGPS)
unit. There is good agreement between the GX5's magnetometer-derived measurement and the DGPS (not shown). The
Windcube's motion-correction procedure uses its internal IMU for roll and pitch data and the DGPS for the yaw measurements
to correct the 1-second winds. The final 10-minute-averaged results that appear in the STA files were obtained from averaging
these 1 Hz data.

**3.2.1 Reprocessing**

Our approach involved reprocessing the uncorrected wind profiles using attitude data from the GX5 in place of the
Windcube's internal IMU. We started with the uncorrected wind profiles that are stored in the *.stdrtd files. These files contain
the x, y, and z components of the wind field (xwind, ywind, zwind) as measured in the Windcube's frame of reference. These
measurements were obtained from Doppler Beam Swinging (DBS) analysis of individual 5-beam scans (Newman et al., 2016).





Because these results were generated in real-time, the DBS analysis was performed as each new beam came in. Thus, the

*.stdrtd files were updated at the raw beam rate of ~1.0 Hz. This resulted in considerable oversampling, because the true

temporal resolution was determined by the scan time, which for the Windcube was about 5 seconds. Thus, the uncorrected

wind profiles have a true temporal resolution of about 5 seconds but are oversampled at 1-second intervals. We adopted the

same scheme for reprocessing the data to maintain consistency with the *.stdrtd files.


**3.2.1.1 Motion Correction**

Figure 7a and Figure **7**b shows the coordinate system used by the Windcube and its orientation relative to the buoy. The

Windcubes were installed on both buoys with their x-axes pointing bow-ward and y-axes pointing starboard. As a result, the

Windcubes' z-axes are downward. The 3DM-GX5-45's coordinate system is shown in Figure 7c. This device was mounted

upside down on the belly of the lidar with its x-axis coaligned to the lidar's x-axis, i.e., toward the bow. As a result of the

inverted orientation, the y-axis of the 3DM-GX5-45 is pointed toward the port side, and z is up. The relationships between the

Windcube and the GX5 coordinate systems are summarized in Table 5.

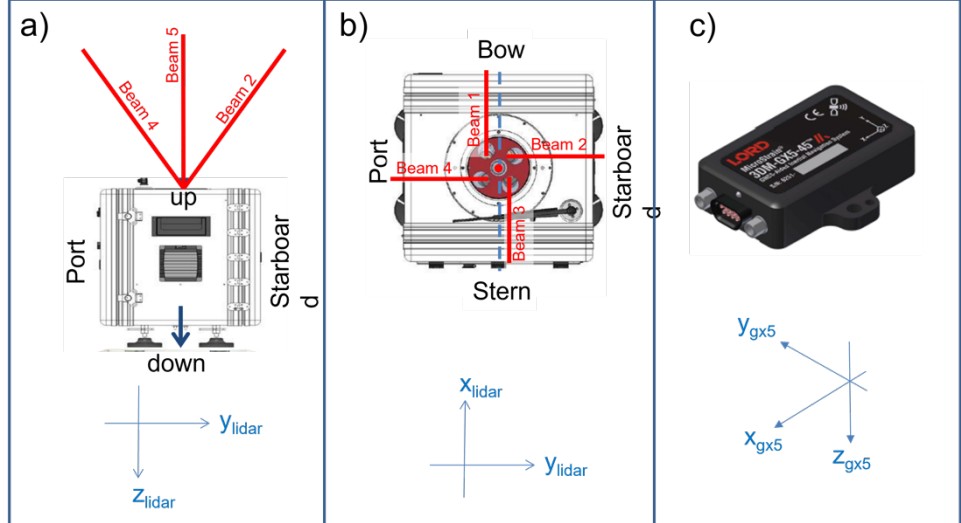

**Figure 7: (a) Side view of the Windcube looking toward the bow, (b) top view of the Windcube, and (c) the 3DM-GX5-45 and its coordinate system. The x-axis of the Windcube points toward the bow, and y points toward the starboard side so that z points down.**

**Table 5. Coordinate systems used by the Windcube and the 3DM-GX5-45 based on their installed positions during the Humboldt and Morro Bay deployments. The 3DM-GX5-45 provides measurements of the Euler angles necessary to transform from platform to Earth coordinates.**

| | IMU | 3DM-GX5-45 |
| --- | --- | --- |



| Orientation | Windcube | platform | Earth |
|---|---|---|---|
| x-axis | bow | bow | North |
| y-axis | starboard | port | West |
| z-axis | down | up | Zenith |

The uncorrected 1-second winds use the coordinate convention listed under "Windcube" in Table 5. The variables called

"xwind", "ywind", and "zwind" in the *.stdrtd files correspond to the bow, starboard, and down directions, respectively. Transforming the Windcube's velocity measurements to the GX5 platform coordinate system simply involves taking the negatives of the Windcube's y and z velocity components, i.e.,

$$\begin{pmatrix} v_{bow} \\ v_{port} \\ v_{up} \end{pmatrix} = \begin{pmatrix} xwind \\ -ywind \\ -zwind \end{pmatrix}.$$
(1)


Each 1-second profile is transformed from Windcube coordinates to an Earth-fixed coordinate system using

$$\begin{pmatrix} v_{north} \\ v_{west} \\ v_{zenith} \end{pmatrix} = A \begin{pmatrix} v_{bow} \\ v_{port} \\ v_{up} \end{pmatrix}$$
(2)

where $A$ is the matrix that transforms a vector from platform coordinates (in bow, port, up) to Earth coordinates (in north, west, zenith). $A$ is given by

$$A = R_3(\alpha)R_2(\beta)R_1(\gamma)$$
(3)

where $\alpha$, $\beta$, and $\gamma$ are the yaw, pitch, and roll angles, respectively. The individual rotation matrices are given by

$$R_1(\gamma) = \begin{pmatrix} 1 & 0 & 0 \\ 0 & \cos\gamma & \sin\gamma \\ 0 & -\sin\gamma & \cos\gamma \end{pmatrix},$$
(4)



$$R_2(\beta) = \begin{pmatrix} \cos\beta & 0 & -\sin\beta \\ 0 & 1 & 0 \\ \sin\beta & 0 & \cos\beta \end{pmatrix} \tag{5}$$


and

$$R_3(\alpha) = \begin{pmatrix} \cos\alpha & -\sin\alpha & 0 \\ \sin\alpha & \cos\alpha & 0 \\ 0 & 0 & 1 \end{pmatrix}. \tag{6}$$


During reprocessing, we first computed the mean pitch, roll, and yaw from the GX5 over the pulse integration time of each lidar beam. This put the GX5 data on the same time grid as the lidar. We noted that the DBS analysis did not account for the variation in the platform attitude during the 5-second period it takes to complete one scan. Instead, we ignored motions with

timescales shorter than the scan duration. Thus, for a given 5-second scan, the roll, pitch, and yaw were further averaged to produce the $\alpha$, $\beta$, and $\gamma$ values used in the transformation matrix, equation (3). We note that in practice, there can be significant platform motion over the scan duration. Obviously, this will cause some error in the results. Averaging the 1-second data helps mitigate noise in the first-order moments (e.g., vertical velocity), but estimation of the second-order moments (e.g., vertical velocity variance) can be problematic. The final post-processed results are averaged like the STA files (Krishnamurthy and

Sheridan et al., 2023b, 2023d).

### 3.3 Wave observations data processing and filtering

Calculations of sea-surface gravity waves (henceforth waves) are derived from analysing pitch, heave, and yaw over a 20-minute time window in frequency space. A 20-minute time window is the industry standard for wave measurements in the United States (NDBC 1996). The 20-minute sampling interval results in less wave data points than the rest of the instruments,

which are sampled at 10-minute intervals. Wave measurements are all derived from the TRIAXYS sensor, and the data were subjected to quality controls to identify spurious data by comparing measurements with adjacent ones and removing values of significant wave height exceeding 40 m (AXYS 2012).

Like the surface measurements, only data within the watch circle were considered good. Thus, data marked as good or

questionable outside of the watch circle by the first processing routines were marked as bad (Krishnamurthy and Sheridan et al., 2023a, 2023c). Data from neighbouring NDBC buoys were used as auxiliary to cross-verify the lidar buoy measurements. Stations 46028 and 46022 are located 7.7 km and 25 km from the Morro Bay and Humboldt deployments, respectively. In addition, these buoys are deployed at depths of 1154 m (46028) and 419 m (46022), which are similar water depths to the lidar



buoys and located far from prominent coastline features that would influence the waves. A wave climate similarity assessment

was performed by pairing data flagged as good from the lidar buoys with NDBC data (the results are shown in Figure 8). Linear regression shows slopes of 0.99 and 0.96 for the Morro Bay and Humboldt deployments, respectively—therefore, both buoy pairs experienced similar wave climates. The minimum significant wave height measured during the deployment period at buoys 46028 and 46022 was 0.71 m and 0.55 m, respectively. Therefore, significant wave heights less than 25 cm were flagged as unrealistic. A fourth level of filtering was applied to peak wave periods and peak wave direction based on significant

wave height. If significant wave height was identified as questionable, then the peak wave periods and peak wave direction were as well. This only applies to questionable data because bad data stems from sensor failure. Finally, if any of the variables were flagged as bad, then all were flagged as bad for that time period because they were all derived from the same sensor. Summaries of the total number of current measurements with each quality flag are summarized in Table 6 and Table 7.

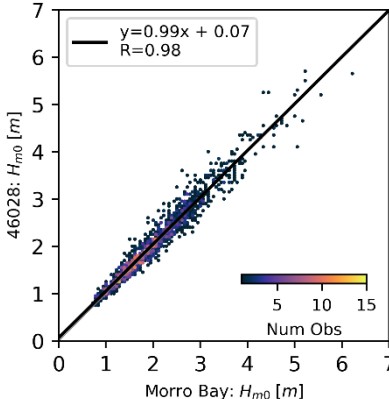
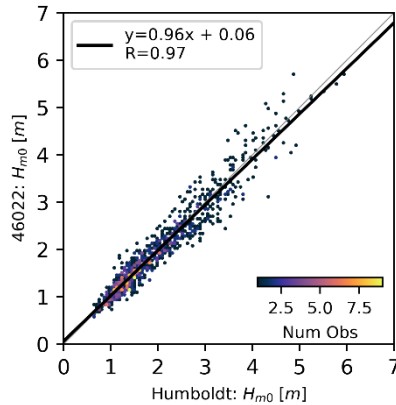


**Figure 8: Significant wave height comparison between (left) Morro Bay and 46028 and between (right) Humboldt and 46022. Linear regression obtained using the least-squares method is shown in each figure along with the Pearson correlation coefficient. The colour scales indicate the number of observations.**

The spread between maximum wave height ($H_{max}$) and significant wave height ($H_s$) was also analyzed to investigate suspect

data. Wave heights have been shown to follow a Rayleigh distribution from which the maximum wave height in a record is estimated as $0.07\sqrt{\ln(N)}\,H_s$, where N is the number of measured waves. During the Morro Bay deployment, the maximum and average $H_{max}$ / $H_s$ were 2.5 and 1.6, respectively, when including questionable data, and were 2.2 and 1.6 when considering good data only. Based on theory, the expected values are 1.7 and 1.6 when including questionable data, and 1.7 and 1.6 when considering only good data. This indicates that the data follows the expected theory. In Morro Bay, only seven

values from previously marked good data exceeded the 2.0 threshold—which has historically been used as a criterion for rogue waves—and have thus been marked as suspect. In Humboldt, 18 points matched this criterion. None of these waves are the largest on the record—therefore, analyses based on extreme waves were unaffected.





Wave peak spread, wave duration, maximum wave period, and maximum wave crest were not included in the b0 file because the instrument does not have the capability to measure them. Finally, the spectral peak wave period and peak wave direction were computed from the wave spectrum. The wave spectrum was estimated using the maximum entropy method (Nwogu 1989) with the TRIAXYS post-processing software version 5.01. The spectral peak wave period is defined as the vertex of a parabola fitted to the maximum discrete period and its two adjacent measurements in the directionally integrated spectrum. Peak wave direction follows the same procedure but with the frequency-integrated spectrum. These two variables do not contain data in the a0 file. Directions in the b0 file represent the direction where waves are coming from, measured clockwise from magnetic north.



**Table 6. Data quality flags at the Morro Bay deployment.**

| Deployment | Morro Bay | | | |
|---|---|---|---|---|
| **Filter** <br><br> **Parameters** | **1** <br> **Bad Data** | **2** <br> **Watch Circle Filter** | **3** <br> **Minimum Energy Flag** | **4** **Good** <br> **Significant** <br> **Wave Height** |
| Average wave height | 149 | 236 | 17 | N/A |
| Average wave period | 149 | 236 | 17 | N/A |
| Maximum wave height | 150 | 236 | 17 | N/A |
| Mean wave direction | 149 | 236 | 17 | N/A |
| Mean wave period | 149 | 236 | 17 | N/A |
| Mean wave spread | 149 | 236 | 17 | N/A |
| Num zero crossings | 170 | 234 | 17 | N/A |
| 10th percentile wave height | 140 | 236 | 17 | N/A |
| 10th percentile wave period | 149 | 236 | 17 | N/A |
| Significant wave height | 149 | 236 | 17 | N/A |
| Significant wave period | 149 | 236 | 17 | N/A |
| Peak wave direction | 149 | 236 | 17 | N/A |
| Peak wave period | 149 | 236 | 17 | Q: 21,276 |
| Spectral max wave height | 149 | 236 | 17 | N/A |
| Spectral peak wave period | 149 | 236 | 17 | Q: 21,038 |




**Table 7. Data quality flags at Humboldt deployment.**

| Deployment | Humboldt | | | |
|---|---|---|---|---|
| Filter<br><br>Parameters | 1<br><br>Bad Data | 2<br>Watch Circle Filter | 3<br>Minimum Energy Flag | 4          Good<br>Significant<br>Wave Height |
| Average wave height | 19 | 2,738 | 34 | N/A |
| Average wave period | 19 | 2,738 | 34 | N/A |
| Maximum wave height | 75 | 2,738 | 34 | N/A |
| Mean wave direction | 19 | 2,738 | 34 | N/A |
| Mean wave period | 19 | 2,738 | 34 | N/A |
| Mean wave spread | 19 | 2,738 | 34 | N/A |
| Num zero crossings | 300 | 2,586 | 34 | N/A |
| 10th percentile wave height | 26 | 2,738 | 34 | N/A |
| 10th percentile wave period | 19 | 2,738 | 34 | N/A |
| Significant wave height | 19 | 2,738 | 34 | N/A |
| Significant wave period | 19 | 2,738 | 34 | N/A |
| Peak wave direction | 19 | 2,738 | 34 | N/A |
| Peak wave period | 19 | 2,738 | 34 | Q: 27,632 |
| Spectral max wave height | 19 | 2,738 | 34 | N/A |
| Spectral peak wave period | 19 | 2,738 | 34 | N/A |

## 3.4    Ocean current and CTD data processing and filtering

All the ocean current data were derived from the Nortek/Signature 250 Acoustic Doppler Currents Profiler (ADCP) in a 10-minute window and, as with the other sensors, were separated into three groups based on the data quality (Krishnamurthy and Sheridan et al., 2023a, 2023c). The three groups included good, questionable, and bad data, with corresponding data quality flags of 0, 1, and 2, respectively. Like the filtering process in Section 3.4, watch circle masks were applied first, with only data within the watch circle considered as good. For current speed, data that met any of the following conditions were marked as questionable:

- The vertical shear of the current speed was greater than 0.2 m s$^{-1}$
- Any data that were located between two NaN values in the vertical profile (missing values were marked as NaN)
- Buoys appeared just out of the watch circle once but then returned.

Data that met any of the following conditions were marked as bad:



- Missing data

- Data from the last day before the number of bins in the ADCP changed—which included 6 October 2020 for the Morro Bay deployment and 28 December 2020 for the Humboldt deployment—to account for service visits

- Data from the last day of the deployment, which was 19 October 2021 for the Morro Bay deployment and 7 July 2022 for the Humboldt deployment

- Bursts of current speed that were temporally and spatially (in depth) uncorrelated and occurred only once (i.e., less

than a 10-minute duration)

- Isolated measurement in time (i.e., measurements that did not have at least two consecutive successful events)

- Data measured during the buoy transit (i.e., outside the watch circle).

Summaries of the total number of current measurements with each quality flag are summarized in Table 8 and Table 9. The maximum values among good data at Morro Bay and Humboldt were 2.01 m/s and 1.45 m/s, respectively. The number of bins

in the ADCP was initially set to 23 bins between September 29, 2020, and October 6, 2020, but then was changed to 50 bins by October 7, 2020, at the Morro Bay deployment to improve the resolution of the data. On the contrary for Humboldt, there was an issue observed with the ADCP, so the number of bins were reduced from 50 to 23 after December 28, 2020.

Conductivity data were derived from the Seabird CTD. Sea-surface temperature (SST) data were derived from two sensors,

one from the Seabird CTD measurement and the other from a YSI thermistor. Watch circle flags were also applied to filter the conductivity and SST data. At the Morro Bay deployment, the minimum and maximum SSTs from CTD among the good data were 9.940 °$C$ and 19.149 °$C$, respectively, and 10.002 °$C$ and 20.385 °$C$ from YSI, respectively. In contrast, at the Humboldt deployment, the minimum and maximum SSTs from CTD among the good data were 9.544 °$C$ and 18.300 °$C$, respectively, and 9.524 °$C$ and 18.796 °$C$ from YSI, respectively. Also note that the SSTs during the first 6 hours from 29 July 2021 to 30

July 2021 at the Morro Bay deployment were marked as bad because values from the CTD temperature sensor were unchanged during these periods. Summaries of the total conductivity and SST with each quality flag are also listed in Tables 4 and 5.

**Table 8. Data quality flags at the Morro Bay deployment. The number of missing data points was not included in the second column.**

| Deployment | Morro Bay | | |
|---|---|---|---|
| Parameters | Data Quality Flag | | |
| | 2: Bad Data | 1: Questionable Data | 0: Good Data |
| Current speed | 15208 | 66927 | 1698119 |
| Current direction | 15208 | 66927 | 1698119 |
| Bin spacing | 446 | 0 | 49199 |



| Head depth | 446 | 0 | 49199 |
|---|---|---|---|
| Blanking distance | 446 | 0 | 49199 |
| Conductivity | 335 | 0 | 51242 |
| Sea-surface temperature (CTD) | 335 | 0 | 51242 |
| Sea-surface temperature (YSI) | 622 | 0 | 51242 |


**Table 9: Data quality flags at the Humboldt deployment. The number of missing data points was not included in the second column.**

| Deployment | Humboldt | | |
|---|---|---|---|
| Parameters | Data Quality Flag | | |
| | 2: Bad Data | 1: Questionable Data | 0: Good Data |
| Current speed | 45748 | 153469 | 1185165 |
| Current direction | 45748 | 153469 | 1185165 |
| Bin spacing | 2475 | 0 | 54386 |
| Head depth | 2475 | 0 | 54386 |
| Blanking distance | 2475 | 0 | 54386 |
| Conductivity | 2147 | 0 | 56283 |
| Sea-surface temperature (CTD) | 2147 | 0 | 56283 |
| Sea-surface temperature (YSI) | 3388 | 0 | 56283 |

## 3.5 Pyranometer data processing and filtering

The two lidar research buoys each include a LI-200SA pyranometer (PYR) designed for field measurements of broadband
global solar radiation (GSR). For the first time, the PYR-measured GSR was used here to assess both the presence of clouds
and the "darkness" of the clouds. Our initial assessment was based on well-established methods developed previously for
identification of clear-sky periods (Long and Ackerman, 2000) and cloud optical thickness (COT; Barnard and Long, 2004)
from shortwave broadband data collected over land. It should be mentioned that the COT is a measure of sunlight attenuation
passing through a cloud layer. Thus, the COT can be considered as a quantity for characterizing cloud "darkness"—clouds
with large COT values (> 10) have a "dark" appearance to a ground-based observer. The COT is related to cloud types (e.g.,
Rossow and Schiffer, 1991), which in turn are common markers of both dynamical and thermodynamic states of coupled
atmosphere–ocean systems. Below is an outline of how these methods developed earlier for continental measurements can be
extended to the more challenging coastal conditions.



During clear-sky conditions, identification of such conditions requires high-resolution (1-minute) measurements of the global (or total) solar irradiance and its direct and diffuse components (Long and Ackerman, 2000). In contrast, the PYR-measured GSR has moderate resolution (10 minutes), and the required measurements of its direct and diffuse components are lacking. To address the lack of required inputs, changes of the GSR measured by PYR for a given day were monitored. The algorithm later checks for clear-sky periods that are long enough to allow for the corresponding empirical fitting to the diurnal cycle of

sunlight described comprehensively by Long and Ackerman (2000). For example, a sufficiently large number (> 100) of clear-sky points are required for empirical fitting of high-resolution (1-minute) data (Long and Ackerman, 2000). Here, a limited number (> 50) of clear-sky points was used for the empirical fitting of moderate-resolution (10-minute) data. Coefficients of this fitting obtained for a given clear-sky day were used to estimate a hypothetical clear-sky GSR (Figure 9a) for a nearby cloudy day. The term hypothetical is employed for the GSR that would be measured by PYR during clear-sky conditions for

the considered cloudy day, i.e., an estimate is made of what the clear-sky GSR would be if the clouds were not present. Finally, the estimated values of clear-sky GSR were utilized to (1) calculate COT (Figure 9b) using the method of Barnard and Long (2004) and (2) estimate a temporal cloud mask (Figure 9c) by assuming that a given 10-minute period is cloudy if the corresponding clear-sky GSR noticeably exceeds the PYR-measured GSR (> 10%). For our initial assessment, the selected threshold (10%) is twice as large as the typical error (5%) of the LI-200SA PYR under natural daylight conditions. The

estimated temporal cloud mask can be used to calculate the average cloud amount for a longer period (e.g., 1 hour) as a fraction of cloudy points blowing over the buoy location (Krishnamurthy and Sheridan et al., 2023a, 2023c). Interpretation of the cloud mask should take into consideration the type of cloud present during the measured period. For example, dense fogs and plumes that occur in coastal areas may have optical thicknesses (up to 4) comparable to the COT of optically thin clouds, indicated by the horizontal magenta line in Figure 9b. To distinguish dense fogs and plumes from optically thin clouds, additional

measurements (e.g., lidar) are needed.



Earth System
Science
Data

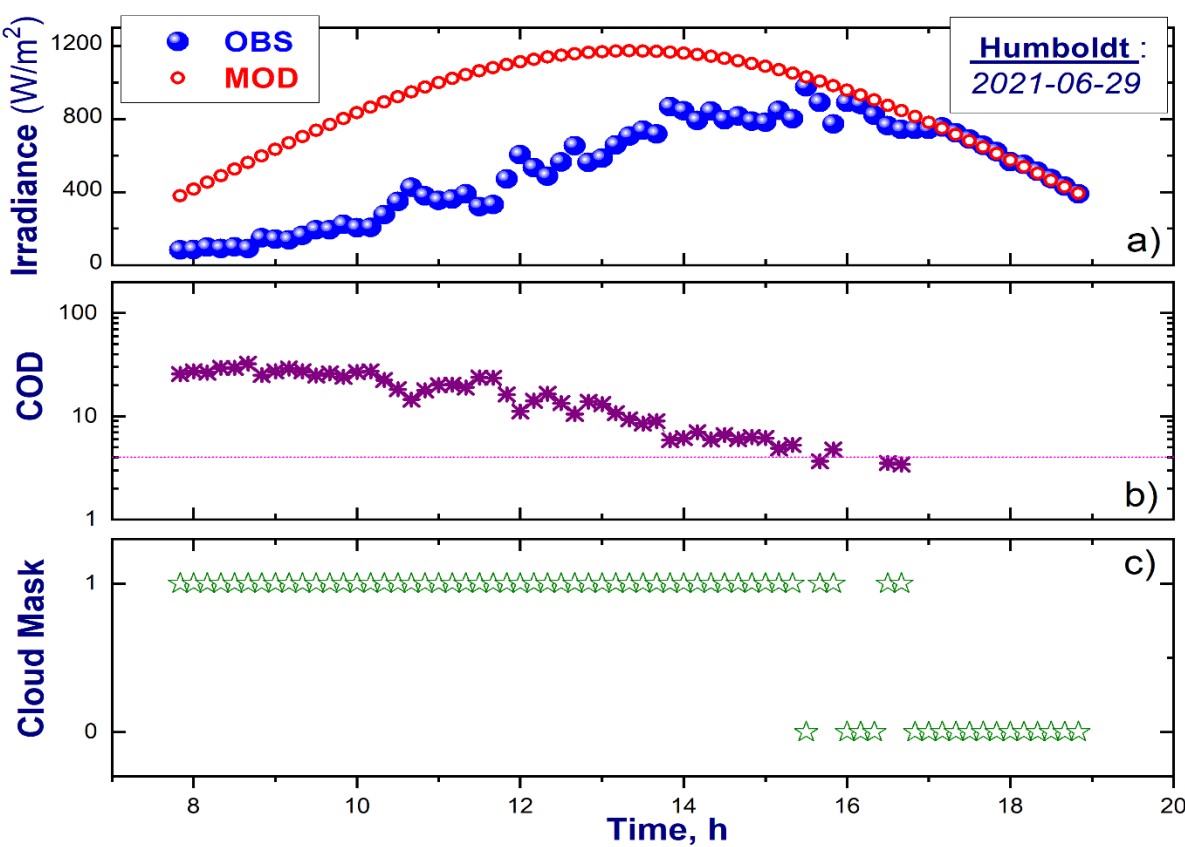

**Figure 9: (a) The GSR measured (OBS) for a given day (2021-06-29) and location (Humboldt) and its estimated (or model) clear-sky (MOD) counterpart, (b) calculated COT, and (c) estimated cloud mask.**

# 4    Results

## 4.1    Surface wind and direction statistics

The two wind speed instruments aboard the buoys were in near-perfect agreement with each other during the Morro Bay and Humboldt deployments, with Pearson's correlation coefficients of 0.9996 and 0.9995, respectively (Figure 10). Additionally, the near-surface wind speeds from the buoy deployments were also in good agreement with wind speed measurements from

the nearest NDBC buoys during the deployment time periods, with correlations of 0.98 between the Morro Bay and NDBC 46028 buoys and of 0.91 between the Humboldt and NDBC 46022 buoys (Figure 10).


**Figure 10: Onboard cup versus sonic near-surface wind speeds (left) and sonic versus nearest NDBC buoy near-surface wind speeds (right) at Morro Bay (top) and Humboldt (bottom).**

The 10-minute averaged near-surface wind speeds have distinct seasonal and diurnal trends (Figure 11). At Morro Bay, the fastest wind speeds, as averaged by hour of day over the month, occurred during the late spring, with a maximum of 9.17 m s$^{-1}$ in the month of May, and the slowest wind speeds occurred during the summer, with a minimum of 4.58 m s$^{-1}$ in the month of August. Diurnally, the fastest near-surface wind speeds occurred between 00-06 UTC, which corresponds with the evening transition in local time (16-22 Pacific Standard Time).

At Humboldt, a similar seasonal pattern to Morro Bay was observed with some of the fastest near-surface wind speeds occurring in the spring and the slowest occurring in late summer, with the distinct exception of the month of July. The fastest



near-surface wind speeds at Humboldt, 8.14 m s⁻¹ on average, occurred during July, largely driven by the winds between 08-15 UTC corresponding with the middle of the night in local time (00-07 Pacific Standard Time).

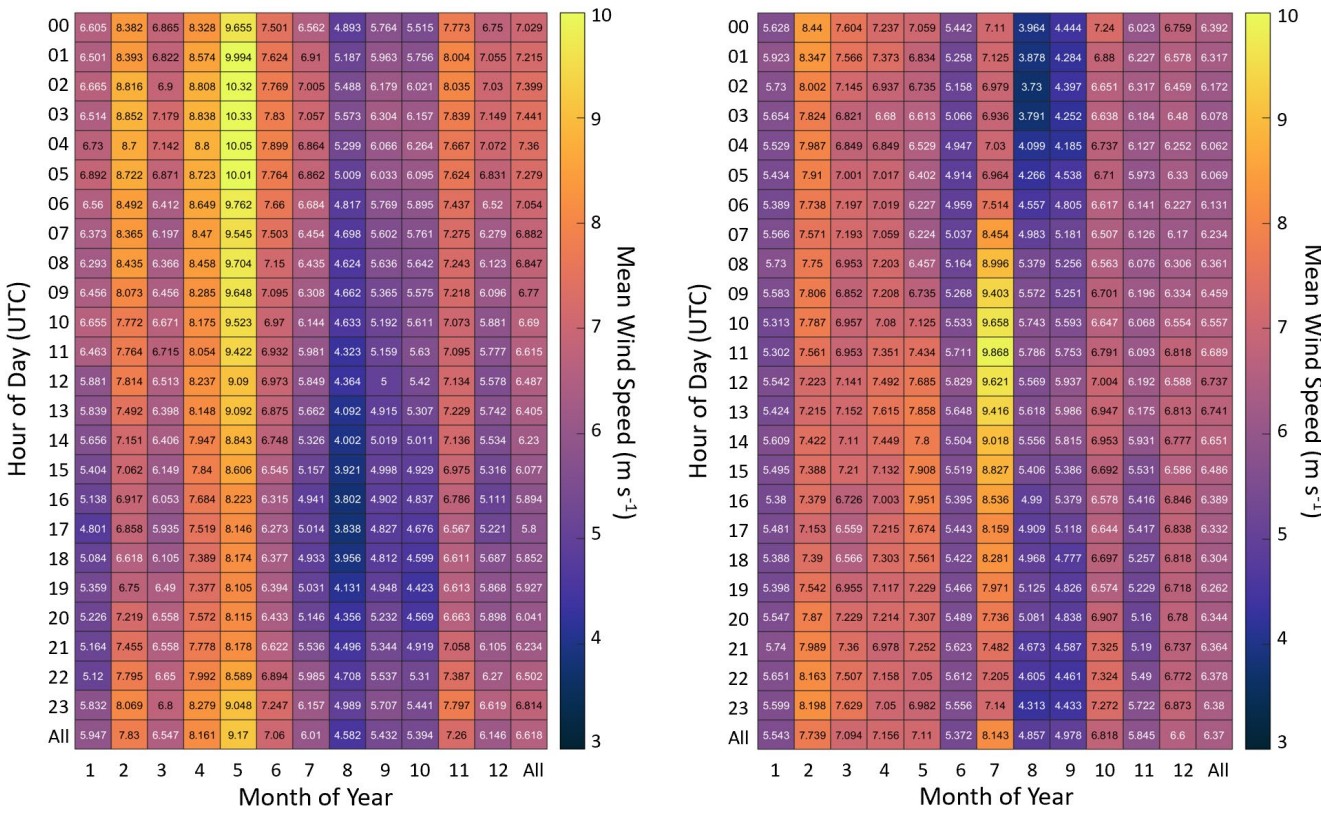

**Figure 11: Seasonal and diurnal average 4 m wind speeds at Morro Bay (left) and Humboldt (right).**

The near-surface wind direction distributions are predominantly uniform for each deployment, with the bulk of wind sourcing from the northwest at Morro Bay and the north-northwest at Humboldt (Figure 12). Wind reversals are observed to occur along the United States Pacific Coast (Bond et al., 1996), and infrequent occurrences of south-easterly near-surface flow were measured by the Morro Bay buoy, characterized by slow wind speeds. At Humboldt, more frequent occurrences of south-south-easterly flow were measured with a greater distribution of wind speeds during the events.



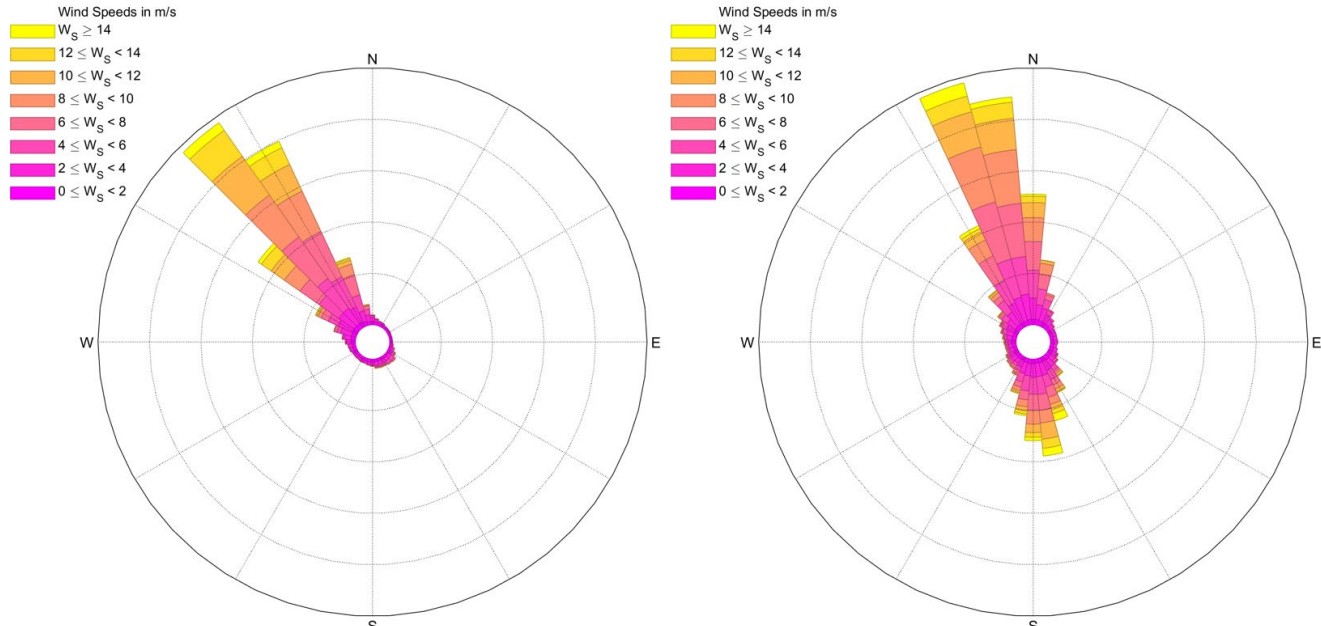

**Figure 12: 4 m wind roses at Morro Bay (left) and Humboldt (right).**

Air and sea-surface temperature show a distinct seasonal trend at Morro Bay (Figure 13). Both temperatures are highest during

the late summer and early autumn and lower in the spring. Temperatures at Humboldt also show a seasonal trend with a warmer

summer and autumn and a cooler winter. The air–sea temperature difference ($\Delta T$), in conjunction with wind speed, has been

shown to be a good predictor for many processes in operational meteorology such as fog incidence and surface heat fluxes

(e.g., Kettle 2015). Performance of reanalysis models has been shown to correlate with atmospheric stability in the region

(Sheridan et al. 2022). In Morro Bay, SST is on average higher than air temperature throughout the year (Figure 13). Negative

$\Delta T$ suggests higher likelihood of unstable atmospheric conditions at the site. In Humboldt, conditions often tend to be stable

in the summer and unstable in the winter.

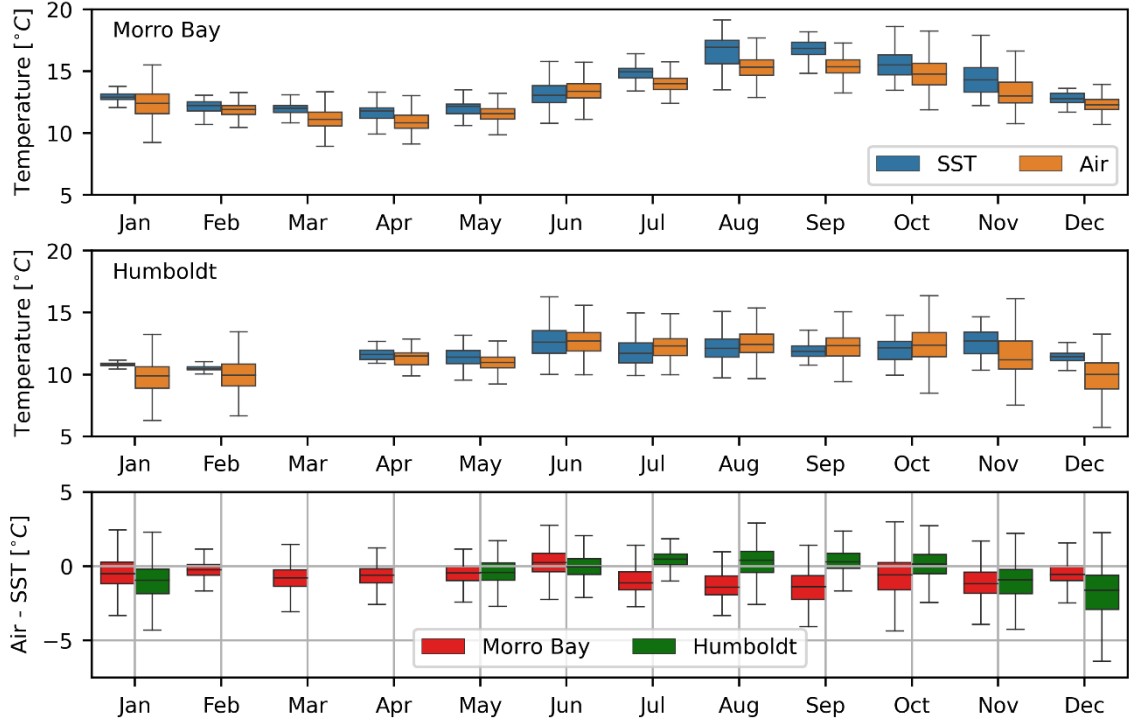

**Figure 13: SST and air temperature at Morro Bay (top) and Humboldt (middle). Air–sea-surface temperature difference (bottom).**
**Data are shown for months with at least 2 weeks of data.**

## 4.2    Doppler lidar wind, direction, and turbulence statistics

Ten-minute averages of wind speed, wind speed variance, wind direction, vertical velocity, and vertical velocity variance were computed from the corrected and uncorrected 1-second wind profiles. The 1-second data were quality controlled by assigning missing values to wind measurements with carrier-to-noise-ratios (CNRs) below –23 dB. Within each 10-minute interval,
variances were computed by first linearly detrending the 1-second data (Krishnamurthy and Sheridan et al., 2023b, 2023d). The data availability was also computed as the percentage of 1-second samples above the CNR threshold (-23 dB). No smoothing or interpolation (in height or time) was applied. To evaluate the impact of the motion-correction procedure, the effect on the median wind speed, turbulence intensity, and vertical velocity profiles was examined. Specifically, the (motion) corrected, uncorrected, and STA results were compared against one another. All the results shown in this section were obtained
by averaging the 10-minute data over the entire deployment. For a given deployment, only those time-height bins with valid samples in all three datasets were used. This ensures that the median profiles are computed under identical meteorological conditions.

Figure 14 shows a comparison between corrected, uncorrected, and STA median wind speed profiles. Motion correction had a very modest effect on the wind speed when compared to the uncorrected winds. The STA wind speeds, on the



other hand, were generally smaller than either the corrected or the uncorrected wind speeds below 150 m and larger above 150

m. For both Humboldt and Morro Bay, there was good agreement between all three results at 160 m, but the corrected and

uncorrected wind speed profiles quickly diverged from the STA profile, indicating stronger wind speed shear in the corrected

(or uncorrected) result between 40 m and 60 m ASL, particularly at the Morro Bay deployment. Overall, we found that the

uncorrected wind speeds were between 0.01% and 0.04% larger than the corrected wind speeds at both Morro Bay and

Humboldt. By contrast, we found the corrected wind speeds to be up to 3.7% larger than the STA winds at Humboldt, and up

to 3.0% larger at Morro Bay.

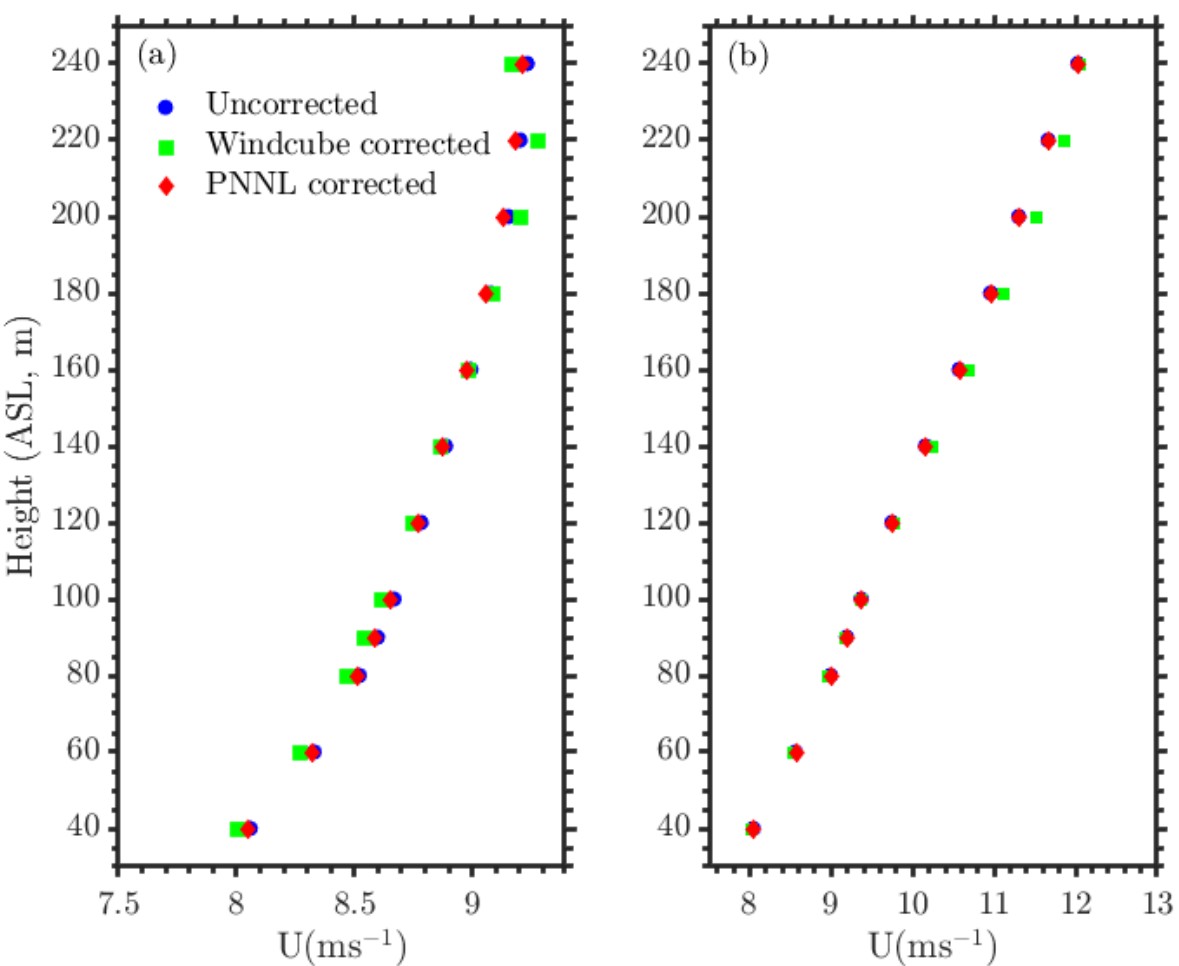

**Figure 14: Impact of motion correction on estimates of the median wind speed profile for (a) Humboldt, and (b) Morro**
**Bay. Motion-corrected results are shown in red, uncorrected results in blue, and the Windcube STA output in green.**

Figure 15 shows a comparison between corrected, uncorrected, and STA turbulence intensity (TI) profiles. TI is defined as the

standard deviation of wind speed over 10-minutes divided by the average wind speed over the same duration. The motion-

corrected result showed the lowest wind speed variance, with the uncorrected variance only slightly larger. In contrast, the
STA TI, were significantly larger than either the corrected or uncorrected TI. For both Humboldt and Morro Bay, these
differences increased with height. Overall, the uncorrected TI were about 0.6% larger than the corrected variances at Humboldt,
and 0.4% larger at Morro Bay. By contrast, the STA TI were 55% higher than the corrected TI at Humboldt and 54% higher
at Morro Bay.

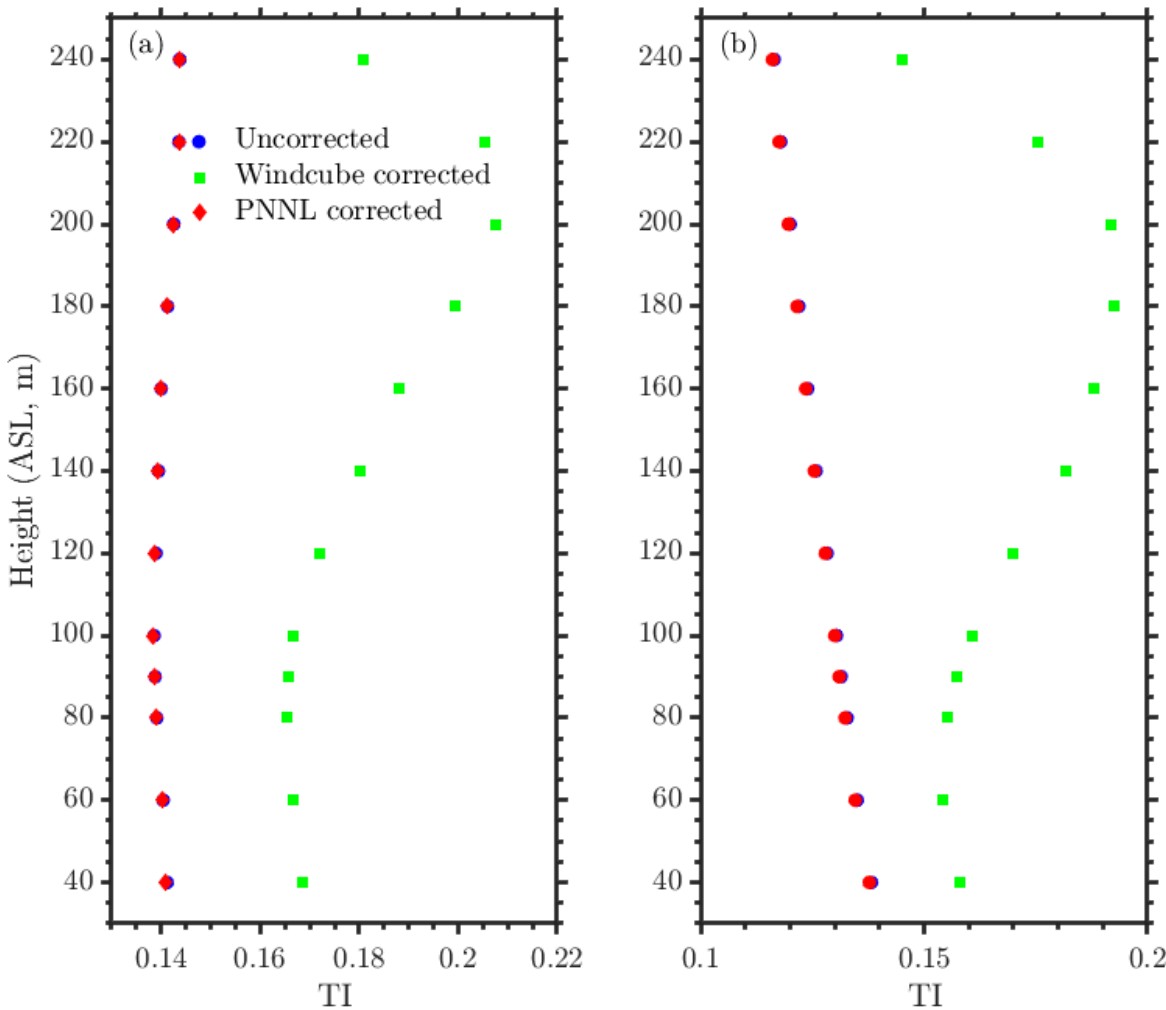


**Figure 15: Comparison of mean turbulence intensity profiles for (a) Morro Bay and (b) Humboldt. The corrected, uncorrected, and STA results are shown in red, blue, and green, respectively.**

Figure 16 shows a comparison between corrected, uncorrected, and STA vertical velocity profiles. Motion correction has a
significant effect when compared to the uncorrected vertical velocity estimates. The STA vertical velocities were considerably
larger than either the corrected or the uncorrected values. Humboldt shows higher vertical velocity than Morro Bay. Overall,

the uncorrected variances were 68% higher than the corrected vertical velocities, and the STA vertical velocities were 172% higher than the corrected variances at Humboldt. For Morro Bay, the uncorrected vertical velocities were about 28% higher than the corrected variances, and the STA variances were 124% higher than the corrected vertical velocities.


**Figure 16: Averaged vertical velocity profiles for (a) Humboldt and (b) Morro Bay. The corrected, uncorrected, and STA results are shown in red, blue, and green, respectively.**

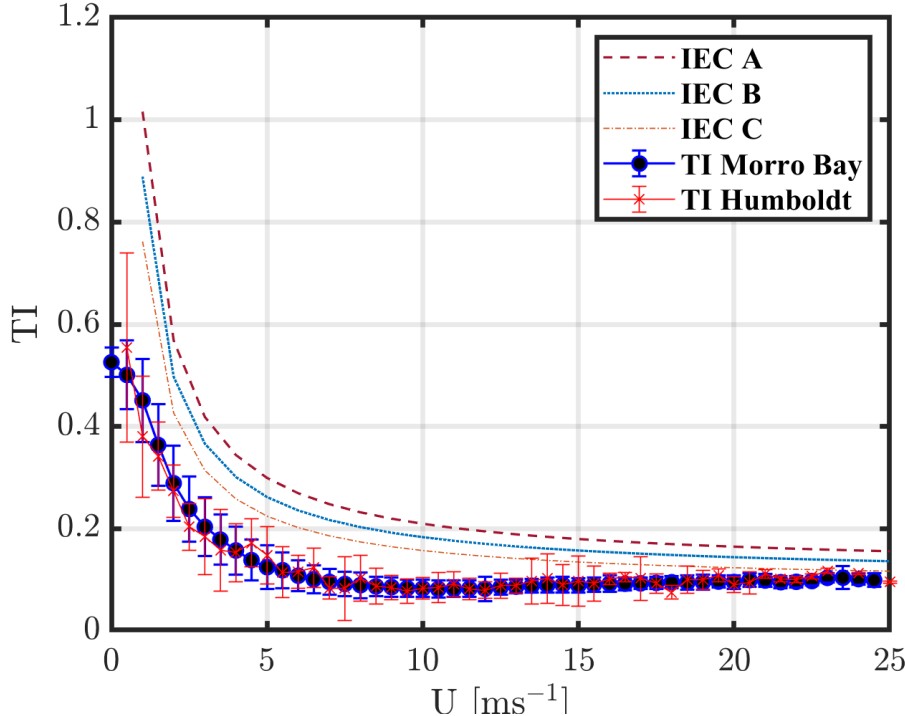


**Figure 17: Bin-averaged turbulence intensity estimates vs. average wind speeds at Morro Bay and Humboldt. The various International Electrotechnical Commission (IEC) wind turbine specific TI curves are also shown.**

The lidar data availability (DA) during the Humboldt and Morro Bay deployments is shown in Figure 18. Missing values in
the uncorrected data are preserved in the corrected data, so the data availabilities for both datasets are the same. The uncorrected dataset uses missing values to flag samples that fall below a predefined CNR threshold. For the Windcube, that threshold is typically set at −23 dB. The STA results incorporate additional quality control that reduces the DA compared to the uncorrected winds. Figure 18 shows that the data availabilities for the corrected/uncorrected winds are significantly higher than those for the STA results at all altitudes. The differences are larger at Morro Bay than they are at Humboldt, and the DA at Humboldt
is slightly lower than at Morro Bay. Overall, the corrected and STA DAs at Humboldt are about 83% and 80%, respectively. At Morro Bay, the corrected and STA DAs are about 92% and 87%, respectively.  Data after December 20, 2021, from the Humboldt deployment is currently under investigation due to an issue observed with the lidar data and is being diagnosed by the vendor.

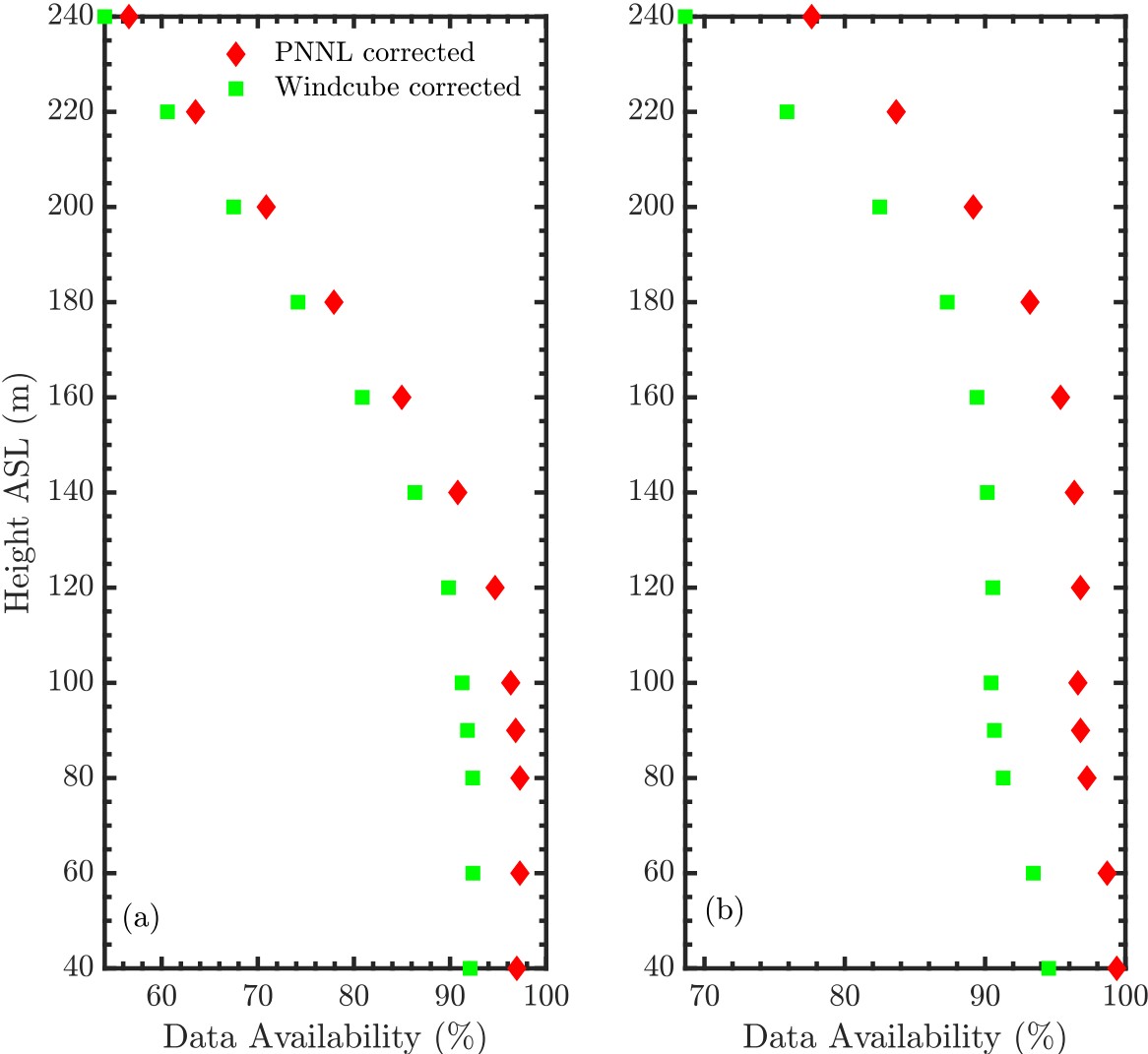

**Figure 18: Comparisons between DA for the corrected winds (red diamonds) and the STA winds (green square). Results are shown for (a) Humboldt and (b) Morro Bay.**

### 4.3 Ocean current and direction statistics

The ocean currents had different spectra at the two buoy deployments, as shown in the rose map (Figure 19). Surface currents were more energetic at the Humboldt deployment than those at the Morro Bay deployment, which were more widely spread. At Morro Bay, the mean and median values of the measured surface current speed were 22.6 cm s⁻¹ and 20.0 cm s⁻¹, respectively, and were 18.6 cm s⁻¹ and 16.0 cm s⁻¹, respectively, for all measured current speeds. Of all currents, 10.7% came from the southeast, and 41.2% of the surface currents came from the northeast toward the southeast. This suggests that the mean kinetic energy peak changed between the surface and greater depths.


At Humboldt, the ocean currents came roughly from the same directions at the surface and at greater depths. For instance, approximately 26.7% of all the currents and 35.3% of the surface currents travelled north to northeast. The mean and median values were 28.1 cm s$^{-1}$ and 26.0 cm s$^{-1}$ for the measured surface current speed, respectively, and 21.8 cm s$^{-1}$ and 20.0 cm s$^{-1}$ for all measured current speeds.


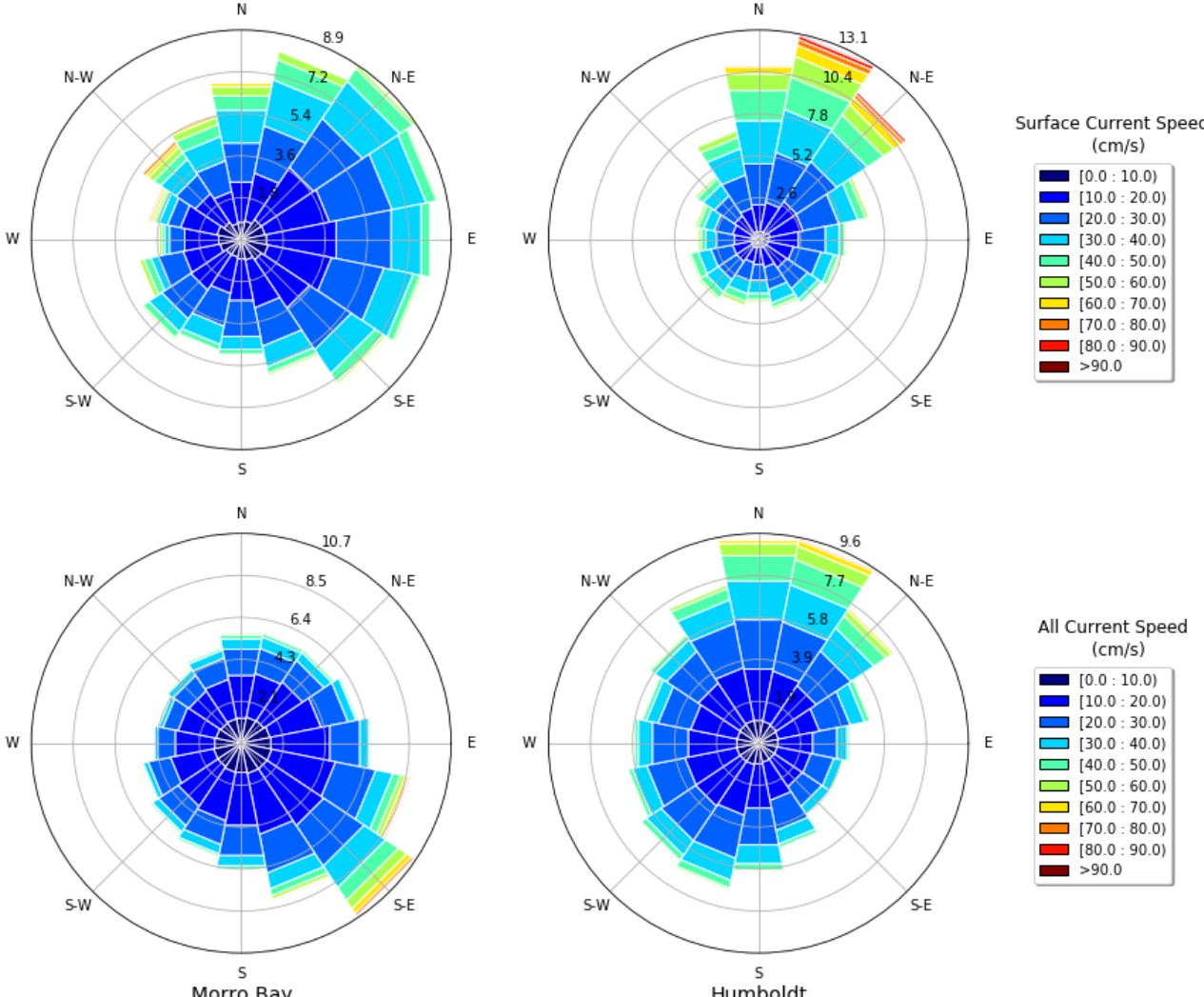

Figure 19: Rose map of ocean currents at the surface (upper) and at all depths (lower) at both Morro Bay (left) and Humboldt (right). Only good data with quality control were included in the analysis.

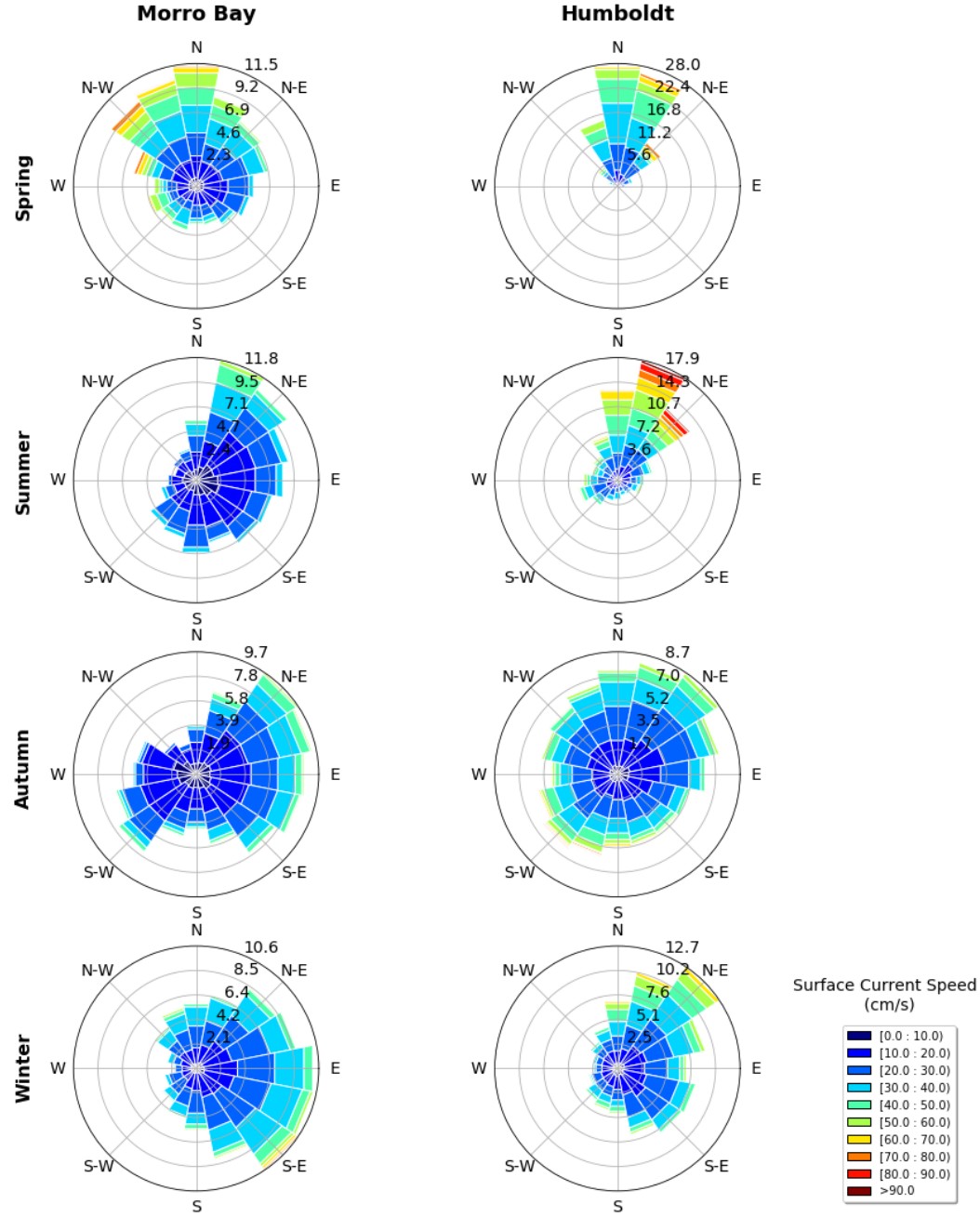

**Figure 20: Rose map of seasonal surface currents at Morro Bay (upper) and Humboldt (lower). Only good data with quality control were included in the analysis.**

There were strong seasonal variations in the surface current at both deployments (Figure 20). The average surface current speed during spring, summer, autumn, and winter at the Morro Bay deployment was 27.2 cm s$^{-1}$, 18.0 cm s$^{-1}$, 19.4 cm s$^{-1}$, and 24.8 cm s$^{-1}$, respectively. At the Humboldt deployment, it was 33.3 cm s$^{-1}$, 33.1 cm s$^{-1}$, 25.0 cm s$^{-1}$, and 25.5 cm s$^{-1}$, respectively.





The mean surface current speed during each season at the Morro Bay deployment was 25.0 cm s$^{-1}$, 16.0 cm s$^{-1}$, 17.0 cm s$^{-1}$, and 24.0 cm s$^{-1}$. At the Humboldt deployment, it was 32.0 cm s$^{-1}$, 30.0 cm s$^{-1}$, 23.0 cm s$^{-1}$, and 24.0 cm s$^{-1}$, respectively. The surface current at Morro Bay predominantly came from the north (11.5%), north-northeast (11.8%), northeast (9.7%), and southeast (10.6%) during each season, respectively. At Humboldt, it predominantly came from the north (28.0%), north-northeast (17.9%), northeast (8.7%), and southeast (12.7%) during each season, respectively. Note that the seasonal

representation of surface currents during winter and spring at the Humboldt deployment may vary because there were no current data available during January–April 2021 and March 2022.

## 4.4    Waves

The wave climate in California, north of Point Conception, is characterized by energetic winters and milder summers (e.g.,

Yang et al. 2020). The monthly distribution of the waves is shown in Figure 21. At Morro Bay, the mean and median significant wave heights during the winter and summer were 2.88 and 2.80 m, and 1.92 and 1.87 m, respectively. At Humboldt, the mean and median significant wave heights during winter and summer were 2.80 and 2.64 m, and 1.93 and 1.75 m, respectively. The spring and autumn, transition periods, have wave heights in between these ranges. Both buoys simultaneously collected data from 8 October 2020 through 28 December 2020 and from 25 May 2021 through 16 October 2021. During that time, waves

measured at Humboldt were more energetic than those at Morro Bay, consistent with the expected longitudinal variability of the wave climate off the California coast.

The full historical record from buoys 46022 and 46028 has also been analysed to contextualize the measurement period. Station 46022 has been active since 1982 and 46028 since 1983, thus providing significant historical records. The historical context

of the measurements can be provided by comparing the full record with the measurements taken at the corresponding NDBC buoys during the time in which the lidar buoy deployments were active. Figure 21 shows box plots at the neighbouring buoys during the full record and the overlapping period. At buoy 46028, the average significant wave height measured during the campaign corresponded to conditions that were more energetic than the long-term average, with 25[th], 50[th], and 75[th] percentiles of 2.25 m, 2.83 m, and 3.50 m vs. 1.9 m, 2.52 m, and 3.26 m, respectively. At 46022, the winter trends were similar with 25[th],

50[th], and 75[th] percentiles of 2.31 m, 3.15 m, and 4.00 m vs. 2.20 m, 2.90 m, and 3.79 m for the measurement period and long-term average, respectively. During the summers, the conditions are also marginally above the long-term average at both stations.  The historical records also show that extreme events with wave heights above 10 m have occurred at these locations. Although such events were not measured during the deployment period, they can be inferred from the neighbouring buoys.



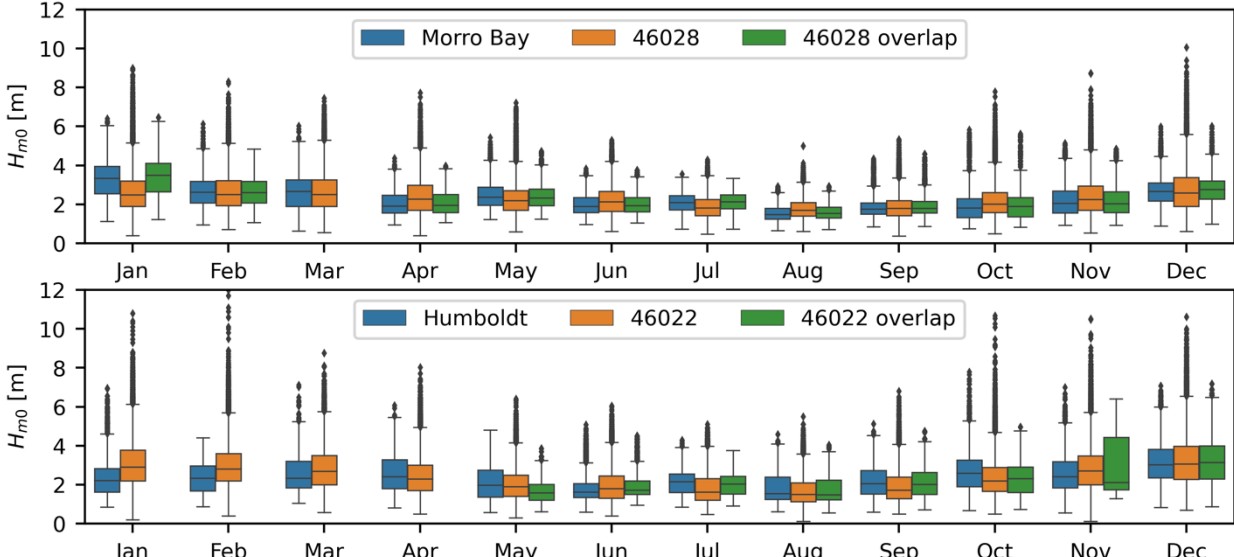

**Figure 21: Monthly average significant wave height distributions at (top) Morro Bay and (bottom) Humboldt. The buoys must have been active for at least 2 weeks for a month to be considered in the analysis.**

Waves near California are also seasonally variable in period and direction (e.g., Villas Bôas et al. 2017), and the lidar buoys measured these cycles at the WEAs (not shown for brevity). In addition, California experiences a multi-modal sea state where multiple sea states approach the coast simultaneously (Villas Bôas et al. 2017, Yang et al. 2020). The presence of multi-modal sea states in the WEAs complicate the description of the surface roughness. Model-based analysis of event prediction in the Mid-Atlantic Bight showed the effect that two-way coupled wave and atmospheric modelling has on accurate prediction of events (Gaudet et al. 2022). In that case, the atmospheric model obtained the surface roughness using the Taylor and Yelland (2001) parameterization with the wave model as input during runtime. Recent results suggest that using the mean wave period in surface roughness parameterizations can provide better results than the peak period (Sauvage et al. 2022). This dataset of collocated wind and wave measurements provides data for validation of these approaches. During the deployments, surface roughness estimates derived from the Taylor and Yelland (2001) parameterization for peak and mean wave periods show a difference of at least one order of magnitude. Figure 22 shows time series of surface roughness from October 2020 through January 2021 at both buoys. The differences were consistent between the buoys.

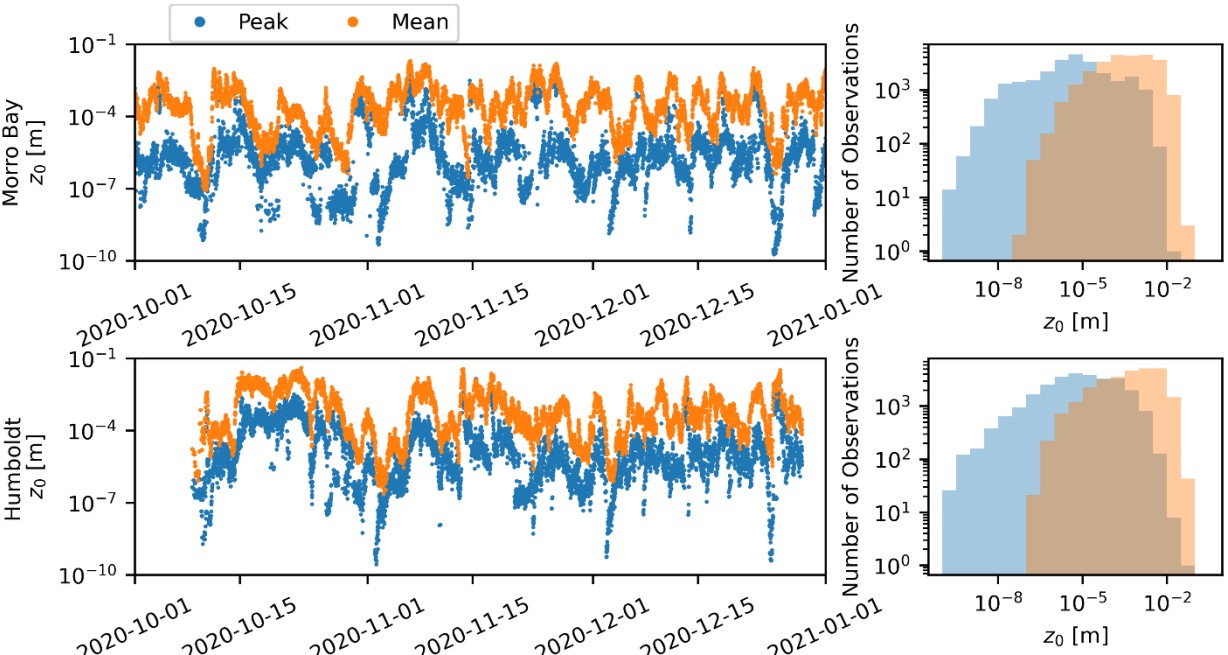

**Figure 22: Surface roughness (top) Morro Bay and (bottom) Humboldt.**

### 4.5 Cloud statistics

Clouds play a critical role in Earth's radiation balance. Atmospheric models have difficulty representing turbulent mixing processes within the boundary layer, which in turn affects the cloud representations in models. Therefore, studying the impact of clouds on boundary layer turbulence and vice-versa is important to improve the accuracy of current-generation weather models. Figure 23 shows hourly cloud fraction estimates from the pyranometer data for the Morro Bay and Humboldt deployments. Both deployments show similar cloud distribution patterns, but the Morro Bay deployment shows significantly higher density of clouds during the summer season compared to Humboldt. Additional analysis on the impact of turbulence due to the presence of clouds is a part of future work.



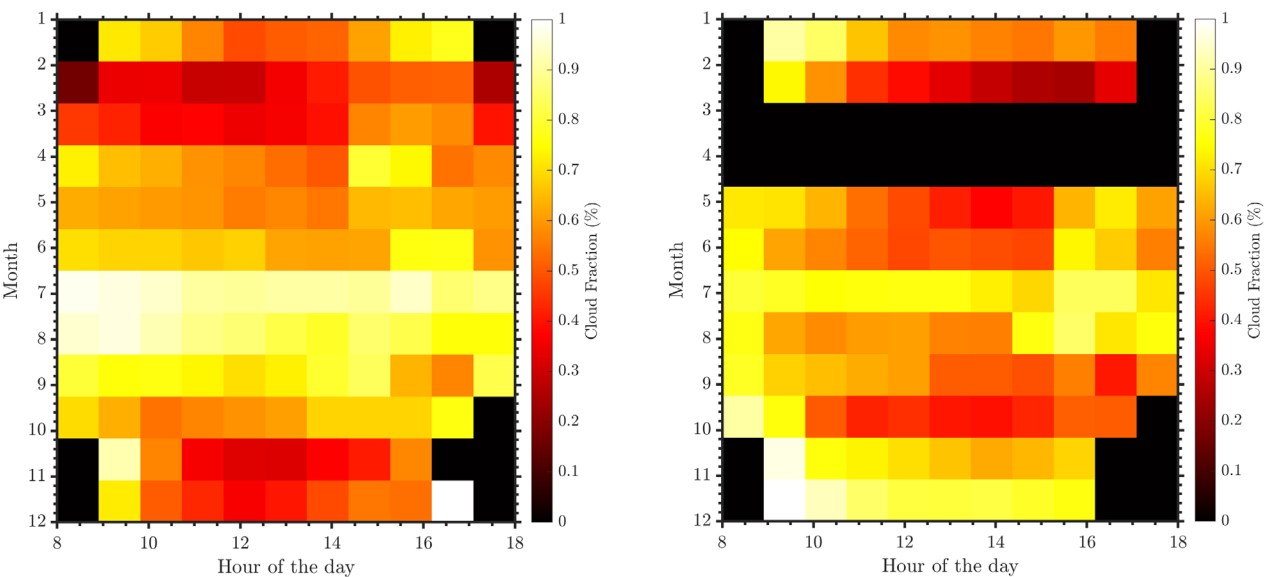


**Figure 23: Cloud fraction estimates for Morro Bay (left) and Humboldt (right). A zero cloud fraction indicates no measurements during that time or data with poor quality.**

## 4.6     Deviations from similarity theory

Theoretical wind profiles based on on Monin Obukhov (MO) similarity theory (MO) are often used in wind energy studies to extrapolate surface or near-hub-height measurements to hub height or above. During homogeneous and stationary atmospheric conditions, the non-dimensional wind shear (Φm), per MO similarity theory, is a function of atmospheric stability and is given by Eq. (7)

$$\Phi_m(\eta) = \frac{kz}{u_*}\frac{\partial u}{\partial z} \tag{7}$$

where $z$ is the height, $u$ is the velocity, $\eta = z/L$ is the stability parameter, $L$ is the Obukhov length (Monin and Obukhov 1954), $u_*$ is the friction velocity, and k is von Karman's constant (0.4). The Obukhov length is a function of the friction velocity and buoyancy flux. Integrating Eq. (7) between $z_o$ (surface roughness) and $z$ (surface layer height) yields the well-known logarithmic wind profile equation

$$\bar{U}(z) = \frac{u_*}{k}\left[\ln\left(\frac{z}{z_o}\right) - \Psi_m\left(\frac{z}{L}\right)\right] \tag{8}$$

where $\Psi_m$ accounts for the influence of stability on the wind profile (Monin and Obukhov 1954). The Obukhov length ($L$) is given by



$$L = -\frac{u_*^3 \overline{\theta_v}}{kg\left(\overline{w'\theta'_v}\right)_s} \tag{9}$$

where $\theta_v$ is the virtual potential temperature, $g$ is the gravitational constant, and $\left(\overline{w'\theta'_v}\right)_s$ is the surface virtual potential temperature flux. When turbulent fluxes are not directly measured, they can be estimated from a bulk method when near-surface measurements of non-turbulent quantities are available (Fairall et al. 1996, Fairall et al. 2003, Edson et al. 2013). The bulk method—the Coupled Ocean–Atmosphere Response Experiment (COARE)—relies on vertically integrated forms of MO similarity equations to relate interfacial differences of temperature, moisture, and wind components to their vertical turbulent

fluxes. Necessary inputs include near-surface air temperature, relative humidity, wind speed, pressure (to compute air density), SST, significant wave height, and the phase speed of dominant waves. In general, the resultant bulk flux relations cannot be solved for the turbulent fluxes in closed form because the Obukhov's length is itself a function of those fluxes. Thus, the bulk flux algorithm (except in idealized cases) becomes an iterative method to find a self-consistent set of turbulent fluxes for given non-turbulent inputs.


Deviations of MO theory within the marine boundary layer have been observed during stable atmospheric conditions (Vickers and Mahrt 1999), low wind and fast swell cases (Grachev and Fairall 2001), internal boundary layers (Vickers and Mahrt 1999), and within the wave boundary layer (Davidson 1974, Donelan et al., 1993, Smith et al., 1992). Most of these studies were conducted closer to the coast and observed larger wave-induced stresses, which affected the surface layer wind

profile. In deep waters several kilometres away from the coast, the applicability of MO theory for non-dimensional wind shear estimates has still not been established in neutral and stable atmospheric conditions. In unstable atmospheric conditions, observations of wind shear follow conventional non-dimensional wind shear forms (Edson and Fairall 1998, Edson et al., 2004), although the measurements were made at higher elevations compared to typical surface buoy heights. MO similarity theory is valid only within the surface layer, which is typically assumed to be equal to 10% of the atmospheric boundary layer

depth (*h*, Stull 1998). Shallow marine atmospheric boundary layers can be observed offshore, which would significantly limit the application of similarity theory for wind energy applications. Figure 24 shows wind shear estimates measured between the surface anemometer and lidar measurements at 40 m as a function of atmospheric stability at the Morro Bay and Humboldt locations. The observed wind shear estimates were closer to the formulations from Beljaars and Holtslag 1991. Appendix A provides more details on the various similarity theory formulations used in Figure 24.


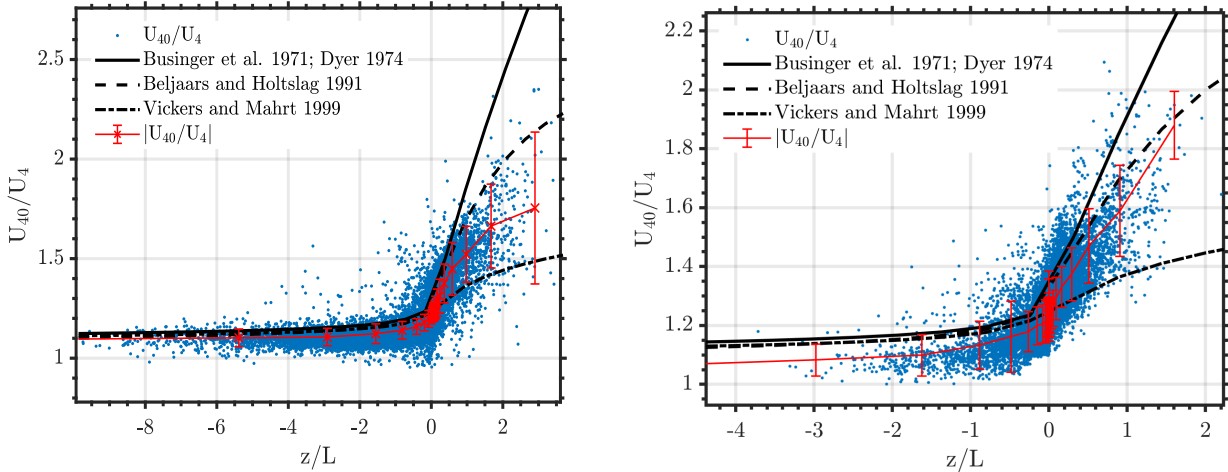

**Figure 24: Wind shear observations at Morro Bay (left) and Humboldt (right) as a function of atmospheric stability. The blue dots represent 10-minute observations of shear, the solid red line with crosses represents the bin-wise average, the solid black line represents the Bussinger-Dyer model wind shear estimate, the dashed black line represents the Beljaars and Holtslag model wind shear estimate, and the dash-dotted line represents the Vickers and Mahrt model wind shear estimate.**

## 5    Code and data availability

All raw and post-processed data from this study are currently available on the Wind Data Hub website, https://a2e.energy.gov/data#. The raw data (*.00) are used to create the final post-processed files (*.b0). Near-surface, wave, current, and cloud datasets for Humboldt are provided at https://a2e.energy.gov/ds/buoy/buoy.z05.b0 (DOI: 10.21947/1783807; Krishnamurthy, R., and Sheridan, L. (2023b)), and for Morro Bay at https://a2e.energy.gov/ds/buoy/buoy.z06.b0 (DOI: 10.21947/1959715; Krishnamurthy, R., and Sheridan, L. (2023a)). Lidar datasets for Humboldt are provided at https://a2e.energy.gov/ds/buoy/lidar.z05.b0 (DOI: 10.21947/1783809; Krishnamurthy, R., and Sheridan, L. (2023d)) and for Morro Bay at https://a2e.energy.gov/ds/buoy/lidar.z06.b0 (DOI: 10.21947/1959721; Krishnamurthy, R., and Sheridan, L. (2023c)). Additional codes to read the raw lidar data are available at https://github.com/rkpnnl/DOE_Buoy_DAP. All post-processed data are available in standard NetCDF or csv format.

## 6    Conclusions

PNNL, in partnership with DOE and BOEM, deployed two buoys off the coast of California in Fall 2020 in areas targeted for offshore wind development. The buoys were outfitted with state-of-the-art instruments, including Doppler lidars, to further our understanding of atmospheric and oceanographic characteristics of the area and provide much needed data to



inform the siting and leasing of offshore wind energy. Buoy measurements are valuable for studying basic science of offshore wind profiles, validating, or calibrating atmospheric and oceanographic models, and developing new parameterization schemes. In this article, a summary of measurements, data-processing details, and results from lidar and other instruments on the buoys is presented. The final post-processed data is currently available to download on the Wind Data Hub.

All surface meteorological data were filtered and compared with nearby NDBC buoy measurements, with wind speed correlations of 0.98 at Morro Bay and 0.91 at Humboldt. In addition, the buoys provided a sense of the local atmospheric stability in the two regions—there was a higher likelihood of unstable atmospheric conditions at Morro Bay, while conditions were stable in the summer and unstable in the winter at Humboldt. Atmospheric models tend to deviate during stable atmospheric conditions, and such data are valuable for evaluating the accuracy of model simulations (Bodini et al., 2022). Novel analysis using buoy-based pyranometers was also conducted, which provided details about the cloud cover and aerosol optical depth over the regions. Along the U.S. West Coast, shallow marine boundary layers have been frequently observed (Beardsley et al., 1987, Burk and Thompson 1996), which are generally encountered with clouds. Clouds are known to affect boundary layer turbulence, and a thorough understanding of how current-generation atmospheric models predict cloud patterns within the region is therefore important for future research. In addition, thorough analysis of the wave sensor and ocean current data was performed, providing details of the multi-model sea state in the call areas about this essential characteristic for floating offshore wind farms. These data will also help support the development and validation of coupled ocean–wind–wave models (Gaudet et al., 2022).

In the Doppler lidar data, motion correction had a small impact on the 10-minute wind speeds when compared to the uncorrected winds. At Humboldt and Morro Bay, negligible differences were observed between the uncorrected wind speeds and the corrected wind speeds. The STA wind speeds, on the other hand, were found to be about 4% higher than the corrected wind speeds at Humboldt and about 3% higher at Morro Bay. Differences between the corrected and uncorrected results were larger for second-order moments like variance or TI. At Humboldt, the uncorrected TI was on average about 0.6% higher than the corrected result. For Morro Bay, the uncorrected TI was about 0.4% higher than the corrected result. By contrast, STA TI are significantly larger than either the corrected or uncorrected results. STA TI were on average 54% larger than the corrected variances at Humboldt and 55% larger at Morro Bay. Motion correction also impacted estimates of vertical velocity. At Humboldt, the uncorrected vertical velocity was 68% higher compared to the corrected result. For Morro Bay, the uncorrected vertical velocity was 28% higher than the corrected result. By contrast, STA vertical variances were 172% larger than the corrected variances at Humboldt and 124% larger at Morro Bay. For turbulence estimates, the net effect of motion correction is primarily to reduce the horizontal and vertical velocity variances. The STA results presented here were obtained using erroneous pitch and roll information from the Windcube's internal IMU. As a result, the STA results contain unrealistically large estimates of velocity variance that in turn result in unrealistically large estimates of turbulence kinetic energy and TI.

Over the last few decades, logarithmic wind profiles have been used extensively in the wind energy resource assessment studies (e.g., Holtslag, 1984; Emeis, 2010, 2014; Drechsel et al., 2012; Krishnamurthy et al., 2013). In particular, the logarithmic wind profile model has been used to extrapolate observed wind speeds to hub height (e.g., tower measurements), interpolate





winds between two atmospheric model levels (Sheridan et al., 2020), and extrapolate geostrophic winds to hub height using the friction velocity computed from the geostrophic-drag law (Tennekes, 1973). Because these models typically break down above the surface layer, such models must be used with caution during shallow marine boundary layers. Using the lidar data collected over an annual cycle, it was observed that the similarity theory model developed by Beljaars and Holtslag (Beljaars and Holtslag 740 1991) compared well with observations. Other models tend to either overestimate or underestimate the shear within the region.

The analyses contained in this article provides significant new information about the offshore conditions along the U.S. West Coast. In addition, the experience gained will inform both configurations and analysis of the data from future deployments of these lidar buoy systems.



Appendix A: Monin Obukhov Similarity theory

Three prominent similarity-theory-based models are generally used in atmospheric studies—the Businger-Dyer (BD; Businger et al. 1971; Dyer 1974), Beljaars and Holtslag (BH; Beljaars and Holtslag 1991) and Vickers and Mahrt (VM; Vickers and Mahrt 1999) models. For reference, their stability functions are given here. The BD functions for stable ($\eta \geq 0$) and unstable atmospheric ($\eta < 0$) conditions are given by

$$\psi_{BD}(\eta) = \begin{cases} -6\eta, \text{ for } \eta \geq 0 \\ 2\log\left(\frac{1+x}{2}\right) + \log\frac{1+x^2}{2} - 2\operatorname{atan}(x) + \frac{\pi}{2}, \text{ for } \eta < 0 \end{cases} \tag{A1}$$

where $x = (1 - 19.3\eta)^{1/4}$. Similarly, the BH stability functions are given by

$$\psi_{BH}(\eta) = \begin{cases} -a\eta - b\left(\eta - \left(\frac{c}{d}\right)\right)exp(-d\eta) - bc/d, \text{ for } \eta \geq 0 \\ \frac{3}{2}\log\left(\frac{1+x+x^2}{3}\right) - \sqrt{3}\,atan\left(\frac{2x+1}{\sqrt{3}}\right) + \frac{\pi}{\sqrt{3}}, \text{ for } \eta < 0 \end{cases} \tag{A2}$$

with a = 1, b = 2/3, c = 5, d = 0.35, and $x = (1 - 12.87\eta)^{1/3}$. The VM stability functions are given by

$$\psi_{VM}(\eta) = \begin{cases} -3x + \log\eta - \log(1-x) + \frac{1}{2}\log(x^2 + x + 1) + \sqrt{3}\,atan\left(\frac{2x+1}{\sqrt{3}}\right), \text{ for } \eta \geq 0 \\ 2\log\left(\frac{1+y}{2}\right) + \log\left(\frac{1+y^2}{2}\right) - 2\operatorname{atan}(y) + \frac{\pi}{2}, \text{ for } \eta < 0 \end{cases} \tag{A3}$$

where $x = (1 + 16\eta)^{1/3}$ and $y = (1 - 35\eta)^{1/4}$.

Appendix B: Data Files, Naming Convention and List of Instruments

Data collected from past and current buoy deployments are made available for public access within the DAP (https://a2e.energy.gov/data). All processed data are uploaded after complete data sets are recovered from the buoy during schedule maintenance visits and after buoy recovery. Data available from the buoy processed data files include the measurements described in Table B1.  All times are in UTC.

The file naming convention used for the data files is:

AAAA.z##.b0.yyyymmdd.hhmmss.BBB.a2e.ccc
where:

- AAA is data source:
  - *buoy*



- o *lidar*
- ## is the buoy deployment number. For example,
  - o *05* is for the Humboldt deployment
  - o *06* is for the Morro Bay deployment
- yyyymmdd is the calendar date where the data file begins
- HHMMSS is the time, in UTC, where the data file begins
- BBB is the measurement type (currents, waves, lidar etc.).
- ccc is the file type:
  - o *.csv*
  - o *.nc*

**Table B1. Description of measurements, variables, and units.**

| Variable name | Description of the variable | Units | Filename | DOI |
|---|---|---|---|---|
| Surface Temperature | Sea surface temperature at ~ -1 m below sea surface from CTD | ºC | buoy.zxx.b0.yyyymmdd.HHMMSS.ctd_conductivity_surfacetemp.a2e.csv | 10.21947/1783807 and 10.21947/1959715 |
| Conductivity | Ocean electrical conductivity from CTD | S m$^{-1}$ | | |
| qc_Surface_Temperature | Quality control for sea surface temperature from CTD | Int | | |
| qc_Conductivity | Quality control for conductivity measurements from CTD | Int | | |
| Surface Temperature | Sea surface temperature at ~ -1 m below sea surface from ysi | ºC | buoy.zxx.b0.yyyymmdd.HHMMSS.ysi_surfacetemp.a2e.csv | |
| qc_Surface_Temperature | Quality control for sea surface temperature from CTD | Int | | |
| Dir$i$ | Current direction in bin number $i$ | degrees | buoy.zxx.b0.yyyymmdd.HHMMSS.currents.a2e.csv buoy.zxx.b0.yyyymmdd.HHMMSS.currents.a2e.nc | 10.21947/1783807 and 10.21947/1959715 |
| Vel$i$ | Current velocity in bin number $i$ | mm s$^{-1}$ | | |
| NumberOfBins | Number of bins: number of measurements being taken in vertical profile | -- | | |
| BinSpacing | Bin spacing: vertical distance between each bin | m | | |
| HeadDepth | Head depth: depth of instrument below ocean surface | m | | |
| BlankingDistance | Blanking distance – or the distance between the transducer head and the first measurement | m | | |
| qc_Vel$i$ | Quality control for current velocity measurements | Int | | |
| qc_Dir$i$ | Quality control for current direction measurements | Int | | |



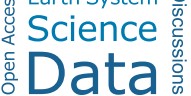

| Variable name | Description of the variable | Units | Filename | DOI |
|---|---|---|---|---|
| gill_wind_speed | Surface horizontal wind velocity, 2D ultrasonic anemometer | m s$^{-1}$ | buoy.zxx.b0.yyyymmdd.HHMMSS.meteo.a2e.csv buoy.zxx.b0.yyyymmdd.HHMMSS.meteo.a2e.nc | 10.21947/1783807 and 10.21947/1959715 |
| gill_wind_direction | Surface horizontal wind direction, 2D ultrasonic anemometer | degrees | | |
| wind_speed | Surface wind speed, cup anemometer | m s$^{-1}$ | | |
| wind_direction | Surface wind direction, wind vane | degrees | | |
| rh | Relative humidity | % | | |
| air_temperature | Air temperature | ºC | | |
| pressure | Atmospheric pressure | mbar | | |
| qc_variables | QC Diagnostic variables | Int | | |
| Column 2 | Measured All-Sky Solar Irradiance | W m$^{-2}$ | buoy.zxx.b0.yyyymmdd.HHMMSS.clouds.a2e.csv | 10.21947/1783807 and 10.21947/1959715 |
| Column 3 | Estimated Clear-Sky Solar Irradiance | W m$^{-2}$ | | |
| Column 4 | Estimated Cloud Optical Depth | Int | | |
| Column 5 | Estimated Cloud Mask. | Int | | |
| | | | | |
| ZCN | Number of zero down crossings | | buoy.zxx.b0.yyyymmdd.HHMMSS.waves.a2e.csv | 10.21947/1783807 and 10.21947/1959715 |
| Hsig | Significant wave height | m | | |
| Havg | Average wave height | m | | |
| Tavg | Average wave period | sec | | |
| Tsig | Significant wave period | sec | | |
| H110 | Wave height, average of highest 1/10$^{th}$ of waves | m | | |
| T110 | Wave period, average of highest 1/10$^{th}$ of waves | sec | | |
| MeanPeriod | Mean wave period | sec | | |
| MeanDirection | Mean wave direction | degrees | | |
| MeanSpread | Mean wave spread | degrees | | |
| PeakPeriod | Mean peak period | sec | | |
| PeakDirection | Peak wave direction | degrees | | |
| qc_variable | Quality control for each variable | Int | | |
| wspd | Wind speed at height $i$ | m s$^{-1}$ | lidar.zxx.b0.yyyymmdd.HHMMSS.sta.a2e.nc | 10.21947/1783809 and 10.21947/1959721 |
| wdir | Wind direction at height $i$ | degrees | | |
| u | Wind component in x-direction at height $i$; horizontal component of wind in the N-S direction | m s$^{-1}$ | | |
| v | Wind component in y-direction at height $i$; transverse component of wind in the E-W direction | m s$^{-1}$ | | |
| w | Wind component in z-direction at height $i$; vertical component of wind | m s$^{-1}$ | | |



| Variable name | Description of the variable | Units | Filename | DOI |
|---|---|---|---|---|
| wspd_var | Windspeed variance at height $i$ over averaging time interval | m s$^{-1}$ | | |
| u_var | Wind variance in x-direction at height $i$; horizontal component of wind in the N-S direction | m s$^{-1}$ | | |
| v_var | Wind variance in y-direction at height $i$; transverse component of wind in the E-W direction | m s$^{-1}$ | | |
| w_var | Wind variance in z-direction at height $i$; vertical component of wind | m s$^{-1}$ | | |
| uv_cov | Horizontal momentum flux (standard deviation) at height $i$ | m s$^{-1}$ | | |
| uw_cov | Streamwise vertical momentum flux at height $i$ | m s$^{-1}$ | | |
| vw_cov | Transverse vertical momentum flux at height $i$ | m s$^{-1}$ | | |
| wspd_raw | Non-motion compensated horizontal wind speed | m s$^{-1}$ | | |
| wspd_raw_var | Non-motion compensated horizontal wind speed variance | m s$^{-1}$ | | |
| wdir_raw | Non-motion compensated horizontal wind direction | degrees | | |
| xwind | Non-motion compensated wind component in x-direction at height $i$; horizontal component of wind in the N-S direction | m s$^{-1}$ | | |
| ywind | Non-motion compensated wind component in y-direction at height $i$; horizontal component of wind in the E-W direction | m s$^{-1}$ | | |
| zwind | Non-motion compensated wind component in z-direction at height $i$; vertical component of wind | m s$^{-1}$ | | |
| xwind_var | Non-motion compensated wind variance in x-direction at height $i$; horizontal component of wind in the N-S direction | m s$^{-1}$ | | |
| ywind_var | Non-motion compensated wind variance in y-direction at height $i$; horizontal component of wind in the E-W direction | m s$^{-1}$ | | |
| zwind_var | Non-motion compensated wind variance in z-direction at height $i$; vertical component of wind | m s$^{-1}$ | | |



| Variable name | Description of the variable | Units | Filename | DOI |
|---|---|---|---|---|
| xwind_ywind_cov | Non-motion compensated horizontal momentum flux (standard deviation) at height $i$ | m s$^{-1}$ | | |
| xwind_zwind_cov | Non-motion compensated streamwise vertical momentum flux at height $i$ | m s$^{-1}$ | | |
| ywind_zwind_cov | Non-motion compensated transverse vertical momentum flux at height $i$ | m s$^{-1}$ | | |
| cnr | Lidar carrier to noise ratio (CNR) at height $i$ | dB | | |
| cnr_var | Minimum lidar CNR at height $i$ over averaging time | dB | | |
| data_availability | Data availability of lidar data at height $i$ over the averaging interval | % | | |
| pitch | Pitch angle from lidar IMU | degrees | | |
| roll | Roll angle from lidar IMU | degrees | | |
| roll_var | Variance of the roll angle from lidar IMU | degrees | | |
| pitch_var | Variance of the pitch angle from lidar IMU | degrees | | |
| lat | Latitude of lidar | degrees | | |
| lat_std | Standard deviation of latitude of lidar | degrees | | |
| lon | Longitude of lidar | degrees | | |
| lon_std | Standard deviation of longitude of lidar | degrees | | |



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

## 8    Acknowledgments

This project was funded by the Wind Energy Technologies Office of the U.S. Department of Energy's Office of Energy
Efficiency and Renewable Energy, under the management of Shannon Davis and Mike Derby.  Pacific Northwest National
Laboratory (PNNL) is operated by Battelle Memorial Institute for the U.S. Department of Energy under Contract DE-AC05-
76RL01830.  PNNL would also like to thank the Wind Data Hub Team, especially Tonya Martin, Chitra Sivaraman, Max
Levin, Matthew McDuff, Kenneth Burk, and Sherman Beus. The team would also like to acknowledge the buoy contractor,
AXYS technologies, for their support maintaining the buoys and for data verification during the deployment.