# Peer review of "Year-long Buoy-Based Observations of the Air–Sea Transition Zone off the U.S. West Coast"

_Earth System Science Data, 2023_

## Referee Comment (RC1)

**Comments to essd-2023-115**

This paper represents an important contribution to the field of experimental offshore wind energy.

The literature review in the introduction is scarce but in the context of a technical study like the present it can be acceptable. Also, several references are added throughout the manuscript. Some statements need additional references, as reported in the specific comments.

The methodology is described exhaustively including calibration and accuracy of the difference sensor.

The description of the physics should be improved as follows:

- The key concept of wave physics (e.g. significant height, spectral behavior) can be briefly explained for the reader that is not familiar with wave dynamics
- The TI from la profiling lidar is well-known to be affected by severe artifacts (probe averaging, cross-contamination) that should be discussed in this context
- The characterization of the lidar statistics as yearly averages in section 4.2 does not provide insight on the 10-minute differences between the different approaches. It is recommended to show the error on the 10-minute statistics as a function (at least) of wind sector and stability.

A main aspect where the paper can be improved is also its clarity and readability. The authors should try to limit the amount of numerical information in the text and summarize results more in tables or plot (see for instance section 4.3 that shows an average of 2 numbers per line). Actually, many details could be omitted to shorten the manuscript. The last point is left at the discretion of the authors. Also, the order in which text and figures present the two sites should be kept the same for the whole manuscript to improve readability.

The above remarks indicate that minor reviews are needed before acceptance.

**Specific comments**

L 27, "The U.S. … by 2030": please add reference

L 29, "…particularly wind speed and direction at hub-height, where offshore turbine blades will be spinning.": this can be rephrased as: "…particularly where offshore turbine blades will be installed" since 1) turbine do not spin only at hub height, 2) the whole rotor vertical span is of interest generally, 3) oceanographic conditions above the sea level sounds odd.

L 33: please specify if the wind turbines slated to be placed in the mentioned areas are of floating or fixed-bottom type.

L 80: "discuss".

Section 2.3: please add the results of the IMU calibration for both buoys and move it to section 2.1.

L 132-134: the 91% availability of the Humboldt buoy does not match Fig. 3 and Table 2.

Figure 3: please plot all timelines for both buoys on the same time axis.

Section 2.3: the acquisition frequency of the IMUs do not match the data in section 2.3.

L 234: "attitude".

Equations 4-6: please expand the A matrix on the same line though a matrix product. Also, please provide a graphic of the yaw, pitch and roll angles signs. Matrix $R_1$ shows a different rotation direction compared to the other two and it would be interesting to understand why.

L 322: "gravity waves" could be mistaken for atmospheric inertial oscillation. Could "sea-surface wave" be clearer to the generic reader?

L 349: please define maximum and significant wave height.

L 353: add reference to the "theory".

L 355: add reference to "historically".

Table 7: why the criterion for good significant wave height does not apply for the spectral peak wave period in this table but is present in Table 6?

L 406: Tables' references look wrong, please check.

Figure 9: "COT" instead of "COD"

Section 4.1: the title could be "Surface wind speed, direction and temperature statistics"

Figure 13: please describe meaning of error bars.

Section 4.2: the title could be "Doppler lidar wind speed, direction and turbulence statistics"

L 409: Please specify that the STA are the Windcube-corrected statistics.

L 530-531: please clarify that $w$ is larger "in magnitude" in the STA data.

L 531-534: the mentioned vertical velocity variances are not shown.

Section 4.2: please provide an explanation of why the TI and vertical velocity variance is higher in the WIndcube data.

Figure 17 is not mentioned in the text.

L 565: Is referred to the direction or is it a spectral consideration?

L 595-594: please add reference.

Figure 21: please describe meaning of error bars and dots.

L 602-603: the wave heights are not always higher than the all-time average, please clarify.

Figure 22: please describe what "Peak" and "Mean" refer to in calculating the roughness.

L 719-720: the comparison of 10-minute data was not shown, yearly averages were provided instead which could be affected by error cancellation.

---

## Author Comment (AC1)

**Reviewer #1:**

This paper represents an important contribution to the field of experimental offshore wind energy. The literature review in the introduction is scarce but in the context of a technical study like the present it can be acceptable. Also, several references are added throughout the manuscript. Some statements need additional references, as reported in the specific comments.

The methodology is described exhaustively including calibration and accuracy of the difference sensor.

The description of the physics should be improved as follows:

- The key concept of wave physics (e.g., significant height, spectral behavior) can be briefly explained for the reader that is not familiar with wave dynamics.
- The TI from la profiling lidar is well-known to be affected by severe artifacts (probe averaging, cross-contamination) that should be discussed in this context.
- The characterization of the lidar statistics as yearly averages in section 4.2 does not provide insight on the 10-minute differences between the different approaches. It is recommended to show the error on the 10-minute statistics as a function (at least) of wind sector and stability.

A main aspect where the paper can be improved is also its clarity and readability. The authors should try to limit the amount of numerical information in the text and summarize results more in tables or plot (see for instance section 4.3 that shows an average of 2 numbers per line). Actually, many details could be omitted to shorten the manuscript. The last point is left at the discretion of the authors. Also, the order in which text and figures present the two sites should be kept the same for the whole manuscript to improve readability.

The above remarks indicate that minor reviews are needed before acceptance.

We thank the reviewer for carefully reading the article and providing constructive feedback. We believe the quality of the article has improved by addressing the reviewer's comments.

With regards to some of the comments above, since this is a data description paper rather than a research article, the authors wanted to be thorough in how the processing was done so it can be reproduced by anyone using the raw data and understand some of the details that goes into data processing of the buoy data. Similarly, we also wanted to showcase some of the analysis that can be done with this type of data to motivate researchers and make this data more accessible. We have addressed some of the questions above in the updated manuscript, for example adding more description about the wave physics where necessary, additional references, and updated plots for the lidar statistics. For specific comments below, the reviewer comments are in black and the authors responses are in blue.

**Specific comments**

L 27, "The U.S. … by 2030": please add reference

We have added a latest reference and made a correction to the goal. It was initially LCOE reduction by 45% by 2030 (U.S. Department of Energy, 2018 Offshore Wind Technologies Market Report) but based on recent research that goal has been moved to 70% by 2035 (Shields et al., 2022).

Reference added:

Shields, M., Beiter, P., & Nunemaker, J. (2022). A Systematic Framework for Projecting the Future Cost of Offshore Wind Energy (No. NREL/TP-5000-81819). National Renewable Energy Lab. (NREL), Golden, CO (United States).

L 29, "…particularly wind speed and direction at hub-height, where offshore turbine blades will be spinning.": this can be rephrased as: "…particularly where offshore turbine blades will be installed" since 1) turbine do not spin only at hub height, 2) the whole rotor vertical span is of interest generally, 3) oceanographic conditions above the sea level sounds odd.

Thanks for the wordsmithing, the word "spinning" was not in reference to hub-height but offshore locations. We have rephrased this sentence as: "Cost reductions are possible with increased offshore data collection, using lidar buoys to better understand simultaneous meteorological and oceanographic conditions, particularly wind speed and direction within the wind turbine rotor layer, where offshore farms will be installed."

We are not sure where we mention "oceanographic conditions above the sea level" in this section or the entire manuscript. Would request the reviewer to provide additional input.

L 33: please specify if the wind turbines slated to be placed in the mentioned areas are of floating or fixed bottom type.

Over California, its currently only floating offshore wind farms. We have added that to the manuscript now.

L 80: "discuss". Section 2.3: please add the results of the IMU calibration for both buoys and move it to section 2.1.

We now show the calibration results at one of the buoy sites in Appendix C, Humboldt, to show that that all IMU's were calibrated and recorded consistent results at the start of the field campaign. But the Windcube internal IMU results were drifting with time and when deployed offshore as noted in Section 3.2. This is primarily to note that it was not a calibration issue, but just a faulty sensor. We have mentioned this in Section 2.2 of the updated manuscript.

L 132-134: the 91% availability of the Humboldt buoy does not match Fig. 3 and Table 2. Figure 3: please plot all timelines for both buoys on the same time axis.

There is a difference between uptime and data availability. Uptime refers to times when the buoy was operational or powered up (Figure 3), but data availability refers to times when the instrument provided good data (Table 2). We have made one clarification with regards to the Humboldt uptime estimate, as that was based on the time frame ignoring when the buoy was completely turned off due to damage to the buoy's power system (high wave event took the wind turbine out).

Since the two deployments were staggered in timeline, the Humboldt campaign ended in June 2022, while the Morro Bay campaign ended in October 2021. Therefore, we would recommend not having them on the same time axis.

Section 2.3: the acquisition frequency of the IMUs do not match the data in section 2.3.

The reviewer is probably referring to GX3-35 and not GX5-45. The GX3-25 has a different acquisition frequency mentioned in Section 2.2, which is 1 Hz, that is on the buoy as a standard IMU. The GX5-45 is the new IMU installed on the buoy, which has an acquisition frequency of 10 Hz as mentioned in both section 2.2 and Section 3.2. The GX3-25 only provides roll, pitch and yaw measurements and not the position/velocity data that is provided by the GX5-45 at 4 Hz. So, we have clarified that in section 2.2 we are only referring to roll, pitch, and yaw measurements in section 2.2. Hope that clarifies any confusion with regards to the various IMU data on the buoy.

L 234: "attitude".

Not sure if the reviewer is asking us to fix a typo here, but we see the word "attitude" spelled correctly in all instances it is being used in the paper.

Equations 4-6: please expand the A matrix on the same line though a matrix product. Also, please provide a graphic of the yaw, pitch and roll angles signs. Matrix $R1$ shows a different rotation direction compared to the other two and it would be interesting to understand why.

We have added the below explanation to provide additional information on the transformation matrix in Line 325 of the updated manuscript and corrected some typos in the Matrices.

"The order and direction of the rotations described by equations (4) through (6) follow standard aerospace conventions as described in the GX5's documentation (Lord, 2019). When transforming from platform to Earth coordinates, a positive roll value results in port side up and starboard down, which corresponds to a. right-handed rotation about the positive x-axis. For the inverted GX5, a positive pitch corresponds to bow down and stern up, i.e., a right-handed rotation about the positive y-axis (port-ward). Also, for the inverted GX5, a positive yaw corresponds to a counterclockwise rotation of the buoy, i.e., a right-handed rotation about the positive z-axis (upward). In our case, equations (3) - (6) represent the inverse transform from Earth to platform coordinates. As a result,

$R_1(\gamma)$ describes a rotation about the negative x-axis, $R_2(\beta,)$ describes a rotation about the positive y-axis, and $R_3(\alpha)$ describes a rotation about the negative z-axis."

L 322: "gravity waves" could be mistaken for atmospheric inertial oscillation. Could "sea-surface wave" be clearer to the generic reader?

In agreement with the reviewer, we refer to them as "sea-surface gravity waves".

L 349: please define maximum and significant wave height.

Added the definition of significant wave height to line 350. The maximum wave height is defined from the Rayleigh distribution as explained in the text.

L 353: add reference to the "theory".

We interchangeably used "theory" and Rayleigh distribution, leading to confusion. The text now refers to the Rayleigh distribution explicitly.

L 355: add reference to "historically".

References have been added.

Table 7: why the criterion for good significant wave height does not apply for the spectral peak wave period in this table but is present in Table 6?

No measurements were affected by this criterion, the table should have read Q:0 instead of N/A. We have fixed this in this version.

L 406: Tables' references look wrong, please check.

All Table and figure references have been checked in the updated manuscript.

Figure 9: "COT" instead of "COD"

The typo has been corrected. Please see the updated figure below.

[Figure]

Section 4.1: the title could be "Surface wind speed, direction and temperature statistics"

We have updated the title per the reviewer's suggestion.

Figure 13: please describe meaning of error bars.

We have added the following context to the caption of Figure 13 to improve the clarity: "The median monthly temperatures are indicated with the horizontal lines within each box, the 25th and 75th percentiles form the colored box range, and the minimum and maximum temperatures are displayed on the whiskers."

Section 4.2: the title could be "Doppler lidar wind speed, direction and turbulence statistics"

This has been updated as requested.

L 409: Please specify that the STA are the Windcube-corrected statistics.

This was already mentioned in Section 3.2 of the manuscript. Currently in line 239 of the updated manuscript.

L 530-531: please clarify that $w$ is larger "in magnitude" in the STA data.

We have removed the STA data from the manuscript, based on the Reviewer #2's recommendation, given that it was obviously a faulty sensor, and we agree that it was not scientifically useful information.

L 531-534: the mentioned vertical velocity variances are not shown.

That was a typo from an earlier version of the manuscript, we only wanted to show vertical velocities rather than the variances in this manuscript. We have corrected "variances" to "velocities" in the updated manuscript with some revised figures.

Section 4.2: please provide an explanation of why the TI and vertical velocity variance is higher in the WIndcube data.

As shown in Section 3.2, there was a fault in the IMU data recorded on the Windcube. In Figure 5, we show the large deviations in pitch and roll values measured by the internal Windcube IMU and external GX5-45 sensor. Vaisala has since removed their sensor in future versions of the Windcube profiler version for offshore use. We have since then removed the comparison between the Windcube STA results, since that was obviously a faulty IMU and it doesn't add any scientific value showing bad data (as recommended by Reviewer #2).

Additional details are provided in the updated manuscript in lines 267 to 272 and also mentioned below.

"Due to this fault in the Windcube internal IMU data, we observed an increase in turbulence and vertical velocity estimates in the Windcube 1 Hz STA data, as the true lidar beam observations are much closer to the respective beam azimuths and elevation angels than as estimated by the internal IMU data. This artificially induced motion results in overcompensating the 1 Hz data, creating a large error in turbulence estimates. We have observed that these impacts are cancelled in a 10-min averaged wind speed estimate but are amplified when looking at turbulent statistics. Therefore, we recommend not using the Windcube STA files if interested in turbulence estimates from the lidars for these two deployments."

Figure 17 is not mentioned in the text.

We have revised this section in view of some comments from both the reviewers. Figure 19 (previous Figure 17) is currently mentioned in the text.

L 565: Is referred to the direction or is it a spectral consideration?

This refers to the direction of the currents similar to how wind direction is reported. The spectral direction only applies to sea-surface gravity waves.

L 595-594: please add reference.

NDBC has been referenced as the data source for buoys 46022 and 46028 earlier in the document, in Section 1, Section 3, and section 4. The data analysis is original work, and the results are shown in Figure 21.

Figure 21: please describe meaning of error bars and dots.

Figure 21 caption has been updated as follows:

*The median monthly significant wave heights are indicated with the horizontal lines within each box, the 25$^{th}$ and 75$^{th}$ percentiles form the coloured box range, the whiskers are drawn at 1.5 times the interquartile range, and the dots are measurements outside of that range.*

L 602-603: the wave heights are not always higher than the all-time average, please clarify. Figure 22: please describe what "Peak" and "Mean" refer to in calculating the roughness.

Figure 22 caption has been updated as follows:

*Figure 22: Surface roughness (top) Morro Bay and (bottom) Humboldt. Peak and mean indicate surface roughness was calculated based on the peak wave period and mean wave period, respectively.*

L 719-720: the comparison of 10-minute data was not shown, yearly averages were provided instead which could be affected by error cancellation.

We have added some statistics to show the impact of motion compensation on 10-minute averaged data, by showing the error bars which show the 25th to 75th percentile range from the mean.

---

## Author Comment (AC2)

**Reviewer #2:**

This manuscript provides a very thorough technical description of the data collected by the DOE buoy lidars off the California coast. It will serve as an important reference for anyone using both past and future data sets collected by these instruments. It also provides some basic analysis of the observations, including an evaluation of M-O similarity theory, which is interesting. Other than some minor comments below, the paper is ready for publication.

We thank the reviewer for carefully reading the article and providing constructive and positive feedback. We believe the quality of the article has improved by addressing the reviewer's comments. Below the reviewer comments are in black and the authors responses are in blue.

Line 23. What does "statistically averaged data" mean? Is "statistically" necessary?

We agree that statistically is unnecessary and have removed it in the updated manuscript.

Lines 41-42: The use of "1-year" and "annual" in this sentence seem redundant.

Thanks for catching the redundancy. We have fixed that in the updated manuscript to: "So far, to the best of the authors' knowledge, there have been no wind observations collected over an annual cycle within the air-sea transition zone (ASTZ, encompassing the upper oceanic boundary layer and lower marine atmospheric boundary layer, Clayson et al., 2023) off the coast of California."

Line 48: Do the publicly available NYSERDA data pre-date the DOE buoy lidar data?

The NYSERDA buoy data were collected off the U.S. East Coast from approximately 2019 to 2023 and DOE buoy lidar data shown in this article was collected off the U.S. West Coast. So, yes, purely from a timeline perspective, the NYSERDA buoys pre-date the DOE buoy lidar data along the coast of California. Although, the DOE buoys were deployed back in 2015 – 2017 along the U.S. East coast, for more details please see here: https://doi.org/10.2172/1632348.

Lines 223-226. Given all of the problems with the Windcube IMU described here, I don't see the point of even showing any of the wind data derived using that IMU. Unless perhaps the authors are implying that data from all Windcube lidars would have these same problems. But if this is a one-off bad instrument, it doesn't make sense to me to show the data (e.g. figures 14, 15, 16). At a minimum, the authors should state their reason for showing the data if it is from a broken instrument.

The intent was mainly to show that when advanced users download the raw Windcube data, they need to be aware of this issue and we would prefer if they use the *.b0 post-processed data. We ofcourse don't know if this IMU issue is just with the two Windcube's on our buoy or all Windcube's V2.0 but we do know that Vaisala has stopped integrating the IMU in their upgraded version of Windcube's. But we agree with the reviewer that showing faulty data does not provide any scientific value, so we have updated this section currently and mentioned about the IMU issue in the text of the updated manuscript and changed the figures accordingly. We believe this has improved the focus of the article.

Lines 351-354: "During the Morro Bay deployment, the maximum and average Hmax/Hs were 2.5 and 1.6, respectively, when including questionable data, and were 2.2 and 1.6 when considering good data only. Based on theory, the expected values are 1.7 and 1.6 when including questionable data, and 1.7 and 1.6 when considering only good data. This indicates that the data follows the expected theory." I'm not sure I follow this. The difference between 1.7 and 2.2 seems substantial (a 30% difference), so how does this indicate that the data follows expected theory. How far off would it need to be to be considered not to follow theory?

Thank you for bringing our attention to this section. We have further clarified the methodology in the text as some things were not explained in detail. The average $H_{max} / H_s$ indeed follows the expected Rayleigh distribution which is 1.6. We have clarified in the text that we refer to the average when we refer to the expected theory. We did not make the assessment only looking at the maximum as they can be outliers or represent errors in the measurements. The text manuscript now reads:

*This indicates that on average the data follows the expected distribution.*

We now turn our attention to wave heights that exceed the expected $H_{max} / H_s$. There have been verified wave measurements in which the maximum wave exceeds the significant wave height by a factor larger than expected by the Rayleigh distribution. When they do by a factor of 2 (or 2.2 depending on the author) or more they are referred to in the literature as rogue waves. We do not have a way to directly verify the maximum wave height measured in the record. Thus, we decided to use the ratio of 2 (instead of 1.7) to flag waves as suspect. We presented 1.7 as another possible cutoff but is more restrictive than what has been used in other applications.

Figure 9b. The y-axis label says COD, should it be COT?

The typo has been corrected. Please see the updated figure below.

[Figure]

Line 662. "Obukhov length"

The typo has been corrected.